

**Mineral nutrients in Saharan dust and their potential impact on Amazon**
**rainforest ecology**
Joana A. Rizzolo[1], Cybelli G.G. Barbosa[1], Guilherme C. Borillo[1], Ana F.L. Godoi[1],
Rodrigo A.F. Souza[2], Rita V. Andreoli[2], Antonio O. Manzi[3], Marta O. Sá[3], Eliane G.
Alves[3], Christopher Pöhlker[4], Isabella H. Angelis[4], Florian Ditas[4], Jorge Saturno[4], Dan-
iel Moran-Zuloaga[4], Luciana V. Rizzo[5], Nilton E. Rosário[5], Theotonio Pauliquevis[5],
Carlos I. Yamamoto[6], Meinrat O. Andreae[4], Philip E. Taylor[7*], and Ricardo H.M.
Godoi[1**]
*Corresponding Authors: School of Life and Environmental Sciences, Deakin Uni-
versity, Australia. e-mail address: philip.taylor@deakin.edu.au (P.E.Taylor*); Envi-
ronmental Engineering Department, Federal University of Parana, Brazil. e-mail ad-
dress: rhmgodoi@ufpr.br (R.H.M. Godoi**).
[1] Environmental Engineering Department, Federal University of Parana, Curitiba, PR,
Brazil.
[2] State University of Amazonas - UEA, Meteorology Department, Manaus, AM, Brazil.
[3] Instituto Nacional de Pesquisas da Amazônia, Programa de Grande Escala Biosfera
Atmosfera na Amazônia, Manaus, Brasil.
[4] Max Planck Institute for Chemistry, Biogeochemistry Department, Mainz, Germany.
[5] Universidade Federal de São Paulo, Instituto de Ciências Ambientais, Químicas e
Farmacêuticas, Diadema, Brasil.
[6] Chemical Engineering Department, Federal University of Parana, Curitiba, PR, Brazil.



[7] Deakin University, CCMB and CMMR, School of Life and Environmental Sciences,

Geelong, Vic, Australia.

**Abstract**
The intercontinental transport of aerosols from the Sahara is likely to play a significant
role in nutrient cycles in the Amazon rainforest, since it carries many types of minerals
to these otherwise low-fertility lands. Iron is one of the micronutrients essential for
plant growth, and the Amazon rainforest is iron-limited. The main aim of this study was
to assess the input and potential impact of iron bioavailability from Saharan dust, name-
ly, the soluble fraction Fe(II)/Fe(III). Seven other soluble elements that are also essen-
tial for plants were measured. Dust particles entrained in the air were collected and ana-
lyzed, but not dust deposited in rainfall as atmospheric washout. The sampling cam-
paign was carried out at the ATTO site (Amazon Tall Tower Observatory), from March
to April 2015, and samplers were placed both above and below the canopy. Mineral
dust aerosol at ATTO showed peak concentrations for Fe(III) (47.6 ng m$^{-3}$), Fe(II) (16.2
ng m$^{-3}$), Na (470 ng m$^{-3}$), Ca (194 ng m$^{-3}$), K (64.7 ng m$^{-3}$), and Mg (88.8 ng m$^{-3}$) during
the presence of dust transported from the Sahara, as determined by remote ground-based
and satellite sensing data and backward trajectories. Atmospheric transport of weathered
Saharan dust, followed by surface deposition, results in substantial iron bioavailability
across the rainforest canopy. The seasonal deposition of dust rich in soluble iron and
other minerals is likely to affect both bacteria and fungi within the topsoil and on cano-
py surfaces, and especially benefit highly bioabsorbent epiphytes, such as lichens. In
this scenario, Saharan dust can provide essential macronutrients and micronutrients to
plant roots, and also directly to plant leaves. The influence on the ecology of the forest





canopy and topsoil would likely be different from that of nutrients from the weathered
Amazon bedrock, which provides the main source of soluble mineral nutrients.
**Key words:** Amazon forest, Sahara dust, mineral nutrients, bioavailable, soluble iron,
outbreak event, dust transport
**1 Introduction**
The Sahara is the largest source of desert dust to the atmosphere (Ginoux et al.,
2012). Studies are beginning to reveal the extent of the Saharan dust influence on nutri-
ent dynamics and biogeochemical cycling in both oceanic and terrestrial ecosystems in
North Africa and far beyond, due to frequent long-range transport across the Atlantic
Ocean, the Mediterranean Sea and the Red Sea, and on to the Americas, Europe and the
Middle East (Goudie and Middleton, 2001; Hoornaert et al., 2003, Yu et al., 2015; Sal-
vador et al., 2016).
Saharan dust affects climate and atmospheric chemistry at both regional and
global scales. Large scale and mesoscale atmospheric circulation have a key role to play
in the emission and transport of mineral aerosols. Research is ongoing into the effects of
year to year and decade to decade variability of loadings and transport of dust in the
atmosphere (Washington and Todd, 2005).
The Amazon Basin, which contains the world's largest rainforest (Garstang et
al., 1988; Aragão, 2012; Doughty et al., 2015) receives annually about 28 million tons
of African dust each year (Yu et al., 2015). There have been suggestions that Saharan
dust transport across the Atlantic may act as a valuable fertilizer of the Amazon rainfor-
est, providing fundamental nutrients (Swap et al., 1992; Koren et al., 2006; Ben-Ami et
al., 2010; Abouchami et al., 2013). However, little is known about the bioavailability of
these nutrients and their potential affect on rainforest ecology. It is, therefore, important



to understand the source types, source strengths, and the physical and chemical proper-
ties of mineral dust aerosol particles over the Amazon Basin (Guyon et al., 2004).
Plants require many nutrients for healthy development (Marschner, 2012). Iron
(Fe) is an essential micronutrient for plant growth (Morrissey and Guerinot, 2009) and it
is a key element in several important functions and physiological processes. It partici-
pates in chlorophyll function and is required for enzymes critical for photosynthesis,
such as catalase, peroxidase, nitrogenase, and nitrate reductase (Hochmuth, 2011). Plant
bio-functions, such as photosynthesis, respiration and hormonal balance, also require
Fe, along with other elements (Pérez-Sanz et al., 1995).
Under natural soil conditions, Fe(III) occurs bound to minerals, such as hema-
tite, that are not soluble in water (Isaac, 1997; Zhu, 1997), and Fe dissolution is depend-
ent on the water's ligand capacity as well as on than the type or quantity of dust deposit-
ed on the surface (Mendez, 2010).
Two distinct pathways of Fe uptake have been identified in plant roots. Pathway
I, present mainly in dicot plants, reduces Fe(III) to Fe(II) by acidification of the rhizo-
sphere. After this reduction, Fe(II) is transported into cells. In pathway II, compounds
with high affinity for iron are secreted into the rhizosphere, where they react with
Fe(III) and form a chelate complex. This complex is moved into cells by specific trans-
porters (Hell and Stephan, 2003; Morrissey and Guerinot, 2009). In the forest, microor-
ganisms, such as fungi and bacteria, play a role in nutrient cycling, and often employ
multiple distinct iron-uptake systems simultaneously (Philpott, 2006).
Furthermore, Fe-rich dust particles can be transported over long distances and
have considerable time and surface area to take up acids (Shi et al., 2011). An increased
proportion of soluble iron has recently been reported in high altitude Saharan dust com-
pared with ground-based samples (Ravelo-Perez et al., 2016). Thus, particle size, solu-





bility, and bioaccessibility of iron oxides in dust will determine the ultimate influence of
these materials on environmental and biological processes (Reynolds, 2014).
Besides iron uptake, other elements are also essential for plants. Magnesium and
Cu are required for photosynthesis and protein synthesis. Calcium is essential for cell
wall and membrane stabilization, osmoregulation, and as a secondary messenger allow-
ing plants to regulate developmental processes in response to environmental stimuli
(Gruzak, 2001). Zinc is directly involved in the catalytic function of many enzymes, and
with regulatory and structural functions (Broadley et al., 2007). Potassium regulates
osmotic pressures, stomata movement, cell elongation, cytoplasm pH stabilization, en-
zymatic activation, protein synthesis, photosynthesis, and transport of sugars in the
phloem (Kerbauy, 2012).
Atmospheric mineral dust contributes thousands of tons of minerals to tropical
rain forests (Okin et al., 2004; Bristow et al., 2010) and likely contributes to plant nutri-
tion, especially compensating for the poor soils with low inherent fertility (Worobiec et
al., 2007). Amazon lowland rainforest soils are shallow and have almost no soluble
minerals; added to this, heavy rains readily leach soluble nutrients from the ground that
are added from litter decomposition and weathered rocks (Koren et al., 2006).
Saharan desert aerosol can compensate for P leaching from the poor soils of the
Amazon (Gross et al., 2015). Intercontinental transport of dust is likely to be of great
importance to the forest, as it might help to maintain or possibly influence an ecosystem
that has roles in global climate regulation, in addition to maintaining regular rainfall and
storing vast amounts of carbon (Karanasiou et al., 2012). A number of studies have stat-
ed that Saharan dust contributes as a fertilizer to the forest (Swap et al., 1992; Koren,
2006; Bristow et al., 2010; Martin, 2010; Abouchami et. al, 2013; Yu et al., 2015). Oth-



er than for P, the amount of this dust that is available to plants as soluble micronutrients
and macronutrients is unknown, as is the potential influence on forest ecology.
Considering that iron is absorbed by plants only as soluble Fe(II)/Fe(III), it is es-
sential to quantify the intake of this mineral from long-range transported African dust,
and evaluate its potential utilization and effect on the Amazon rainforest as an essential
micronutrient. This research aims to assess the bioavailability of iron, and other ele-
ments in the particulate matter in the Amazon atmosphere transported within African
dust. This is then used to assess the likely effect on rainforest ecology.

**2 Methods**
**2.1 Dust sampling**
Sampling was performed on a 80 m walk-up tower at the Amazon Tall Tower
Observatory (ATTO) site (Andreae et al., 2015), from 19 March to 25 April 2015,
which is within the typical period that dust transport to the Amazon Basin has been ob-
served (Swap et al., 1992; Prospero et al., 2014; Yu et al., 2015). Aerosols were sam-
pled above the canopy at 60 m height and below the canopy at 5 m height, without size
cut-off, and transported in a laminar flow through a 2.5 cm diameter stainless steel tube
into an air-conditioned container. The sample humidity at 60 m height was kept below
40% using a silica dryer. The sample humidity at 5 m height was kept dry with a silica
gel diffusion dryer installed on the inlet line. Atmospheric particles were collected on
Nuclepore® polycarbonate filters at a flow rate of 10 l min$^{-1}$.
The aerosol sampling was performed using the inlet below the canopy at 5 m
height for the first 11 days, and the inlet at 60 m height for the other 26 days. The sam-
ples were collected over 24 or 48 h periods, consecutively, in order to accumulate suffi-
cient mass to be detected by ion chromatography-UV-VIS. After sampling, the filters





were immediately stored in flasks containing nitric acid solution ($HNO_3$ Suprapur) pH
2.0-2.5, in order to interrupt the transition process between the two iron oxidative states
(Fe(II) and Fe(III)) and to stabilize the iron concentrations, according to the methodolo-
gy adapted from Siefert (1998), Bruno et al. (2000), and US-EPA Method 3052 (EPA,

1996).


**2.2 Particle physical properties**
Aerosol particle physical properties were determined at ATTO during the entire
campaign at 60 m height. Mass concentration and particle size distribution were meas-
ured by an Optical Particle Sizer (OPS, TSI model 3330; size range: 0.3–10 μm), sam-
pling every 5 min (Andreae et al., 2015). Equivalent black carbon concentrations ($BC_e$)
were obtained by a MAAP (Multi Angle Absorption Photometer), and the spectral de-
pendency of particle absorption coefficients was determined using a 7-wavelength Ae-
thalometer (Model AE33), both with 1 min resolution. Particle scattering coefficients
were obtained at three wavelengths using an Integrating Nephelometer (Ecotech, model
Aurora 3000). Details of the instrumentation setup are given by Andreae et al. (2015).

**2.3 Determination of mineral aerosol**
Soluble species were determined by ion chromatography (Dionex, ICS-5000) us-
ing conductivity detection for cations and UV-VIS for soluble transition metals. For
cation separation, a capillary column CS12A was used, and for transition metals, a
CS5A column (calibrated to quantify traces of Fe(II) and Fe(III)). Each analysis oc-
curred in triplicate and all measurements were performed from a standard curve injected
under the same conditions as the samples, using Chromeleon® software for processing
the generated chromatograms.






### 2.4 Modeling, remote sensing and meteorological data

174 Air mass backward trajectories were calculated using the Hybrid Single Particle

175 Lagrangian Integrated Trajectory (HYSPLIT) Model from the NOAA Air Resource

176 Laboratory, USA, (National Oceanic and Atmospheric Administration), indicating the

177 airflow toward the ATTO site (Draxler and Rolph, 2015). Thus, dust source areas were

178 inferred by tracking individual dust plumes back to their place of origin (Schepanski et

179 al., 2012) as well as determining transport paths. Trajectories were calculated at three

180 different heights within the atmospheric boundary layer (50, 500, and 1000 m) up to

181 240 h previous. Every 24 h from 19 March to 25 April 2015, a trajectory was calculated

182 with a finishing point at the center of the ATTO site (S 2° 08.752' W 59° 00.335'), at 19

183 h UTC.

184 To analyze Saharan dust outbreak events and transport toward ATTO during the

185 campaign, ground-based and satellite remote sensing products and *in situ* measurements

186 of aerosol particle optical properties were integrated with the atmospheric large-scale

187 wind field. The wind field product was taken from the Modern-ERa Retrospective

188 Analysis (MERRA), a reanalysis data based on the Goddard Earth Observing System

189 Data Assimilation System Version 5 (GEOS-5, Rienecker et al., 2011). Ground-based

190 and satellite remote-sensing aerosol optical properties, namely Aerosol Optical Depth

191 (AOD), were obtained, respectively, from aerosol products of the AErosol RObotic

192 NETwork (AERONET, Holben et al., 1998) and of the Moderate-Resolution Imaging

193 Spectroradiometer (MODIS) aboard the Terra satellite (Remer et al., 2005). Particle

194 optical properties were continuously monitored *in situ* at the ATTO site in 2015 at a

195 height of 55 m. Particle scattering coefficients were measured at three wavelengths us-

196 ing an integrating nephelometer (Ecotech Aurora 3000). Absorption coefficients were



measured at 637 nm using a Multi-Angle Aerosol Photometer (MAAP). Particle single
scattering albedo was calculated based on measured absorption and scattering retrieved
by interpolation at 637 nm. All measurements were taken under dry conditions
(RH<50%).
Micrometeorological data were obtained by sensors installed on the micromete-
orological tower at the ATTO site, at 80 m (Andreae et al., 2015). Daily values were
calculated for vertical wind speed median (W), accumulated precipitation (PRP), and
average air temperature (Tair).
Table 1 shows the sampling frequencies, micrometeorological measurements,
sensors (manufacturers), and sensor heights. For the treatment of high frequency data
(10Hz), computational routines were used. First, the sonic raw data was reduced to 1-
min medians. Subsequently, daily values were calculated.

Table 1. List of instruments installed on the walk-up tower (adapted from Andreae et
al., 2015).

| Sampling frequency | Measurement | Instrument used | Height/Depth (m) | Unit |
|---|---|---|---|---|
| 0.1 s | u, v, w (wind components | 3D ultrasonic anemometer (Windmaster, Gill Instruments Ltd.) | 81.65; 46.0; 36.0 | m/s |
| 60 s | Rainfall | Rain gauge (TB4, Hydrological Services Pty. Ltd.) | 81.0 | mm |
| | Air temperature probe | Thermohygrometer (HMP45C, Vaisala, CS215, Rotronic Measurement Solutions) | 81.65; 40.0; 36.0 | °C |



**2.5 Spore samples**
A Sporewatch spore sampler (Burkard Scientific Pty Ltd, UK) was operated at
80 m height for 24 h on 16 separate days between 28 March and 23 April. Particles



larger than 2 μm diameter were impacted onto an adhesive-coated tape attached to a
drum within the sampler. This tape was removed, mounted onto a microscope slide and
examined with an Olympus BX60 light microscope with brightfield optics. Line scans
were performed to identify fungi, and counts were averaged over 24 h and expressed per
cubic meter of air sampled. All aerosol concentrations are given with respect to air vol-
umes at ambient temperature and pressure.
**3 Results and discussion**
The sampling campaign was performed during a typical period for the occur-
rence of dust transport events in the Amazon forest. Sampling covered a total of 38 days
(19 March– 25 April 2015) and 26 samples of particulate matter were collected.

**3.1 Characterization of particle physical properties**
The mass concentration of particles over the Amazon Basin in the wet season is
typically around 10 μg m$^{-3}$ in locations that are influenced by biomass burning emis-
sions. In the central Amazon, where the influence of biomass burning is less, the mass
concentration is even lower. However, elevated concentrations may occur due to Afri-
can dust events that reach the Amazon forest (Martin et al., 2010). The highest hourly
aerosol concentration recorded during our entire campaign was around 55 μg m$^{-3}$ at the
ATTO site (5 April), with a daily average of 23 μg m$^{-3}$ (Figure 1), and often concentra-
tions were well below 10 μg m$^{-3}$. Previous studies conducted by Worobiec et al. (2007)
at a nearby forest site in Balbina, Amazonia, had also detected an abundance of dust
particles during the same season (23 - 29 March 1998). Artaxo et al. (2013) observed,
only trace levels of P, K, and Zn during the wet season in the central Amazon region.
These were thought to have a biogenic source.






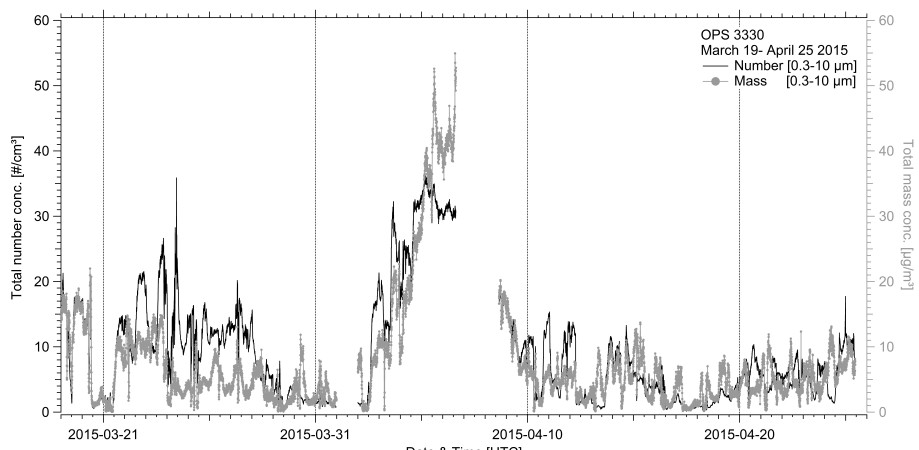


Figure 1. Number concentration (solid lines) and mass concentration (dashed lines) time

series from the OPS instrument (size range: 0.3-10 μm).


For comparison, during Saharan dust events in the Cape Verde archipelago, par-

ticulate matter concentration often exceeds 100 μg m⁻³; a relatively high concentration

when compared to the average aerosol background level of 10-50 μg m⁻³ (Gross et al.,

2015). Obviously, the enhancement in concentrations induced by the plume is highest

near the source, so a larger mass of dust is deposited over the Sahara and the adjacent

Atlantic than over the Amazon rainforest. Notably, the concentrations at ATTO were

still very high in view of the large distance from Africa.

The highest concentrations of black carbon equivalent ($BC_e$) measured online

during this intensive campaign were 0.45 μg m⁻³ (3 April) and 0.37 μg m⁻³ (5 April).

This coincided with the highest mass concentrations of particulate matter. The $BC_e$ con-

centrations were retrieved from light absorption measurements. The particle types that

mostly contribute to light absorption are: combustion generated BC, mineral dust, and

biogenic particles (Moosmüller et al., 2009; Guyon et al., 2004). Therefore, part of the





observed BC could be mineral dust. We further characterized the relative contributions
to $BC_e$ by considering the Absorption Angstrom Exponents (AAE), which reflect the
spectral variability of absorption. Bulk BC particles are expected to have an AAE $\leq 1$
due to increased absorption efficiency at shorter wavelengths (<400 nm). The observed
variability of AAE is further discussed in section 3.3. During the 2008 wet season in the
central Amazon, at a site near Manaus, $BC_e$ concentrations fluctuated between 0.10 and
0.15 µg m$^{-3}$ (Martin et al., 2010). Episodic input of Saharan dust and biomass smoke
transported over long distances from Africa explains the presence of $BC_e$ detected at
ATTO (Andreae et al., 2015).

**3.2 Determination of Mineral Aerosol**
The dominant elements in the soluble fraction of the dust samples were Fe(III),
Zn, Na, K, and Mg (Table 2). The blank fields in the Table correspond to values below
the detection limits, which were calculated according to Method 300.1 USEPA (1997).
The expanded uncertainty (ng m$^{-3}$) was calculated for 95% confidence level, according
to BIPM/GUM (2008).



Table 2. Mineral aerosol characterization of 26 samples collected during the Saharan
dust event that arrived in the Amazon forest during 2015.

| Sampled period (Month/day) | Fe(III) (ng m⁻³) | Fe(II) (ng m⁻³) | Cu (ng m⁻³) | NH₄ (ng m⁻³) | Zn (ng m⁻³) | Na (ng m⁻³) | Ca (ng m⁻³) | K (ng m⁻³) | Mg (ng m⁻³) |
|---|---|---|---|---|---|---|---|---|---|
| 3-19 | 15±0.7 | - | - | 147±4 | 105±8 | 95±7 | 92±13 | 54±12 | 13±2 |
| 3-19 | 11±0.8 | - | - | - | 14±7 | 41±6 | - | 40±8 | 9.0±1.0 |
| 3-20 | 5.6±0.1 | - | - | 163±1.7 | 6.5±3.7 | - | - | - | - |
| 3-21 | 5.8±0.1 | - | - | - | 3.5±1.8 | 84±6 | 40±3 | 26±3 | 13±1 |
| 3-23 | 7.1±0.1 | - | - | 56±2 | 6.4±3 | 73±3 | - | 40±4 | 10.1±0.5 |
| 3-24 | 4.8±0.1 | - | - | 33±1 | 4.2±1.9 | 25±2 | - | 29±2 | 3.0±0.3 |
| 3-25 | 1.8±0.1 | - | - | - | 2.0±0.9 | 44±3 | - | 32±2 | 6.0±0.6 |
| 3-27 | 2.0±0.1 | - | - | - | 3.4±1.6 | 12±1 | - | 22±2 | 1.7±0.2 |
| 3-28 | 1.9±0.2 | - | 0.89±0.87 | 9.7±1.8 | 5.0±2.2 | 18±2 | 5.9±2.5 | 25±3 | 2.6±0.3 |
| 3-30 | 4.1±0.1 | - | - | 5.2±1.3 | 5.8±3.8 | 10±3 | - | - | 1.2±0.5 |
| 3-31 | 4.1±0.1 | - | 2.7±0.8 | - | 4.7±1.9 | 17±2 | - | 8.0±5.4 | 1.7±1.1 |
| 4-02 | 8.5±0.1 | - | 2.5±1.5 | - | 8.3±3.7 | 135±3 | 12±6 | 32±4 | 16±1 |
| 4-03 | 33±0.1 | - | - | - | 4.6±1.8 | 441±4 | 126±4 | 65±2 | 67±1 |
| 4-05 | 48±0.1 | 16±3 | - | - | 8.4±3.6 | 470±4 | 194±4 | 64±4 | 89±1 |
| 4-06 | 33±0.1 | 12±2 | 0.85±0.75 | - | 4.3±1.9 | 220±22 | 128±8 | 44±4 | 49±3 |
| 4-08 | 14±0.2 | 3.3±3.1 | - | 16±1 | 8.9±3.7 | 148±3 | 29±4 | 26±4 | 244±2 |
| 4-09 | 19±1 | 1.6±1.6 | - | 6.6±0.8 | 5.3±1.9 | 57±2 | 29±2 | 15±2 | 8.6±0.3 |
| 4-11 | 5.5±0.1 | - | - | 68±1 | 9.7±3.7 | 84±3 | - | 18±4 | 8.8±0.5 |
| 4-12 | 5.7±0.2 | - | 6.4±0.8 | - | 5.7±1.9 | 38±2 | 5.0±1.9 | 9.6±2.2 | 5.5±0.2 |
| 4-14 | 6.7±0.2 | - | - | 20.4±1.19 | 10.2±3.68 | 24±3 | - | 9.4±4.3 | 3.2±0.5 |
| 4-15 | 12±0.1 | - | 88±1 | 98.5±5.45 | - | 15±2 | - | 7.4±2.05 | 10±1 |
| 4-17 | 7.4±0.2 | - | 9±1 | - | 10.2±3.9 | 24±3 | - | 8.2±4.5 | 3.7±0.5 |
| 4-18 | 1.1±0.1 | - | 2.6±0.8 | - | 4.8±1.7 | 19±2 | 3.5±1.7 | 9.7±2.2 | 2.7±0.2 |
| 4-20 | 1.2±0.2 | - | - | 370±1 | 9.5±3.5 | 29±3 | 8.6±3.5 | 20±4 | 5±0.6 |
| 4-21 | 2.9±0.2 | - | 13±0.7 | 55±1 | 2.1±1.7 | 14±2 | - | - | 4.30±0.2 |
| 4-23 | 2.4±0.1 | - | 1.0±0.8 | - | 2.2±1.9 | 28±2 | - | 5.4±2.2 | 3.1±0.3 |



During the wet season in the central Amazon Basin, Artaxo et al. (2002), Martin

et al. (2010), and Arana and Artaxo (2014), found similar values of K, Fe, Cu and Zn to
those found in our campaign. K, Cu, and Zn are generally considered to be tracer ele-
ments of biogenic emissions from the rainforest, although they also have other sources.
Potassium in submicron aerosols also has a major source from vegetation fires and is
frequently used as a tracer for biomass burning aerosols (Andreae et al., 1983; Martin et
al., 2010). Zhang et al. (2015) studied aerosols from a Chinese tropical rainforest, and
reported that the high abundance of K in fine particles was likely a result of long-range
transport from biomass burning.

Iron, Ti and Al are mainly soil dust related elements (Artaxo et al., 1990; Artaxo

et al., 1994), and are typically present at the highest concentrations during the early wet-
to-dry season transition (February to May), as has been shown in previous studies
(Pauliquevis et al., 2012; Andreae et al., 2015). This is mainly driven by large-scale
atmospheric circulation patterns that favor the transport of dust plumes in a trans-
Atlantic airflow from the Sahara and Sahel regions toward the Amazon basin (Artaxo et
al., 1990; Formenti et al., 2001; Graham et al., 2003; Martin et al., 2010; Baars et al.,
2011; Ben-Ami et al., 2012).

During the wet season, the biogenic aerosol over Amazonia is overprinted peri-

odically by episodes of intense transatlantic transport, which bring Atlantic marine aer-
osols in addition to dust and biomass burning emissions (Bristow et al., 2010; Andreae
et al., 2015). For example, Zhu et al. (1997) studied North African dust entrained in the
trade winds over Barbados (Caribbean) in September, and measured Na concentrations
of 2.4 to 6.5 $\mu$g m$^{-3}$. Barbados is in a region that receives large amounts of Na enriched
marine aerosols due its localization. While these concentrations are higher than those
recorded in the present study at ATTO (220 to 470 ng m$^{-3}$), the co-occurrence of elevat-


ed concentrations of Na and the mineral dust elements, Al, Fe, and Ca, is evidence for
the marine origin of Na in the central Amazon (Talbot et al., 1990).
Artaxo et al. (1990) studied aerosols from the Amazon Basin and noted that the
concentration of total Fe in the fine mode (<2.5 µm) of soil dust were more than 10
times larger in the wet season than in the dry season (101 ng m$^{-3}$ during daytime, 60 ng
m$^{-3}$ during the night and 6.5 ng m$^{-3}$ in the dry season). Pauliquevis et al. (2012) observed
increases in the concentration of total Fe with values reaching 60 ng m$^{-3}$ in the fine
mode mostly during February to April in the Amazon Basin, with a semester average of
36 ng m$^{-3}$. They attributed this to episodes of Saharan dust transport.
In contrast to the high bulk dust concentrations at Barbados, the Fe(II) concen-
trations recorded at ATTO during our sampling campaign (1.6 to 16 µg m$^{-3}$) were sig-
nificantly higher than the Fe(II) concentrations of 0.63 to 8.2 ng m$^{-3}$ measured in miner-
al dust particles collected from the marine atmospheric boundary layer at Barbados (Zhu
et al., 1997). They showed that only a small fraction of the total iron in aerosol particles
was present as Fe(II).
For soluble Fe(III), we found concentrations in the range of 1.10 to 47.6 ng m$^{-3}$
(Figure 2), with the highest concentrations occurring three days in a row (33.7, 47.6 and
32.7 ng m$^{-3}$). This soluble Fe(III) is carried in dust particles that are mainly deposited
onto canopy surfaces by dry sedimentation.  Our soluble Fe(III) concentrations were
significantly higher than those reported by Andreae et al. (2015) from earlier measure-
ments at the same site. They had measured only 1.8 ng m$^{-3}$ of soluble Fe(III) in 120 ng
m$^{-3}$ of total Fe and concluded that the aerosol transport of Fe is not likely to have a sig-
nificant effect on the ecosystem at ATTO.



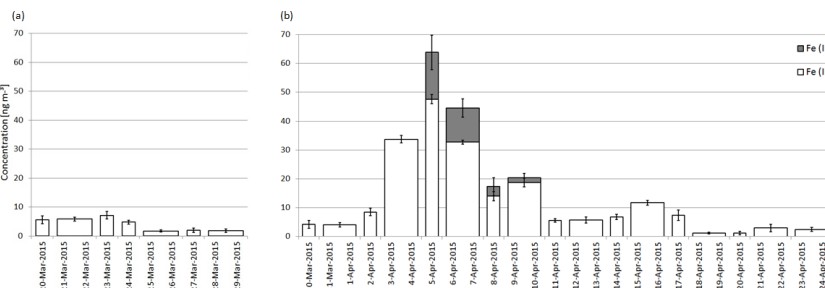

Figure 2. Soluble Fe(III) and Fe(II) concentrations in total particulate matter collected

during the wet season, (a) sampled at 5 m height (20 March to 29 March 2015) and (b)

at 60 m height (30 March to 24 April 2015). The width of bars corresponds to the sam-

pling period: 24 or 48 h.

Desert dust plumes contain iron mainly in the Fe(III) oxidation state, whereas in

industrial effluents Fe is mostly in the Fe(II) oxidation state (Reynolds et al., 2014).

These same authors collected dust from the Sydney area (Australia) and strongly sug-

gested that the addition of Fe(II)-bearing minerals was associated with industrial, urban,

and transportation sources entrained in dust plumes that originally lacked these miner-

als.

Bristow et al. (2010) analyzed aerosol samples collected from the Bodélé De-

pression, Chad, and suggested that the amounts of Fe in some samples likely indicate

the presence of ferromagnesian minerals and also reflect the presence of Fe oxides such

as goethite and hematite, or Fe sulfate salts that have been detected in Saharan dust.

Abouchami et al. (2013), studying the geochemical characteristics of the Bodélé

Depression dust source and the relation with transatlantic dust transport to the Amazon

Basin, found lower Na, K, Fe, and Ca concentrations in Amazon Basin soil samples

than in the Bodélé samples, suggesting that this difference is a reflection of remobiliza-





tion and loss of these elements by chemical weathering under the hot, wet climate con-
ditions in the Amazon Basin.

**3.3 Modeling, Remote Sensing and Meteorological Data**
The largest deposition of iron occurs downwind of the main deserts of the world
- North Africa and the Middle East (Mahowald et al., 2009). Figure 3 (a and b) shows
the backwards trajectories coming not from North Africa nor the Middle East, but in-
stead from the Saharan desert (Formenti et al., 2001; Washington and Todd, 2005; Bris-
tow et al., 2010; Creamean et al., 2013). Koren et al. (2006) estimated that between No-
vember and March, the Bodélé Depression is responsible for most of the dust that is
deposited annually on the Amazon.
Using backward trajectories data (HYSPLIT model), it is possible to observe a
connection between the Sahara and the Amazon. Between 3 and 6 April, the highest
concentrations of Fe(III), Fe(II), Na, Ca, K and Mg in mineral dust samples were ob-
served. According to Figure 3, the air masses arriving in the Amazon during that period
came from the Saharan region.


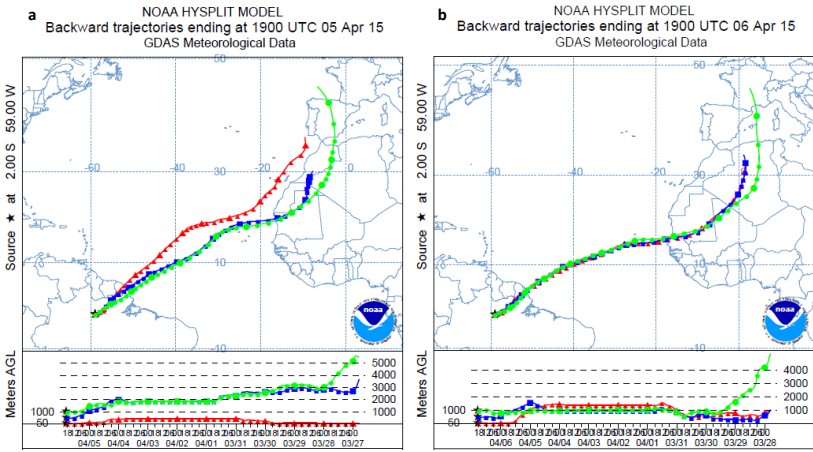



Figure 3. Backward trajectories of air parcels at 50, 500, and 1000 m above the Amazon
for 240 h during the sampling periods in which the greatest concentrations of dust from
the Sahara arrived at ATTO.

High mass concentrations around 3-8 April 2015 coincided with the arrival of

African dust in the Amazon Basin, according to OPS instruments and backward trajec-
tories, respectively, Figure 1 and 3. Periodically in the wet season, long-range transport
of sea spray, Saharan dust, and/or smoke from African biomass burning can deposit
across the Amazon Basin (Martin et al., 2010; Baars et al., 2011; Andreae et al., 2015).

As identified by the AERONET ground based sunphotometers located in Dakar

and in Ilorin, during the campaign period three major Saharan dust outbreaks occurred
and eventually combined with smoke (Figure 4.a). The first outbreak peaked on 22
March and had a stronger effect on AOD over Ilorin compared to Dakar. This feature
was corroborated by the MODIS mean AOD field from 20 to 25 March (Figure 5.a).
During this first event, the atmospheric circulation was not able to promote a significant
transport of the dust and smoke plume towards South America. The influence of Afri-
can particle advection on aerosol optical properties observed at ATTO was weak, but
still detectable. An increase in absorption and scattering coefficients was observed in
comparison to the clean periods (25 March to 2 April and 16-24 April), however, no
significant increase of AAE was observed during this event (Figure 4.b). A less active
dust outbreak from 25 to 30 March followed the first event, as shown by ground-based
and satellite data (Figure 4.a and 5.b).

A second dust outbreak event started at the beginning of April, according to the

AERONET retrievals, and its effects on the African sites extended until 9 April. The
satellite mean AOD field during this period (Figure 5.c) revealed a consistent pattern of



dust transport towards the northeast portion of the Amazon basin, with the wind flow in
the direction of ATTO coming from regions effectively influenced by the Saharan dust
plume. It is possible that smoke from biomass burning in the African sub-Sahel region
joined with dust aerosols transported to the Amazon, since the Ilorin region is affected
by biomass burning emissions in this season (Haywood et al., 2008). Fe(III) and Fe(II)
concentrations in particulate matter increased between 3 and 9 April (Figure 2), and this
correlated with an increase in particle absorption coefficients and a decrease in single
scattering albedo (Figure 4.b and 4.c). The AAE during this event reached values higher
than 5, and after 10 April returned to background levels. The increase in AAE is a
strong indication that dust and/or biomass smoke particles contributed to the observed
increases in absorption coefficients. The correlation between the concentration of crus-
tal elements in particulate matter and aerosol absorption coefficients during African
advection events has also been reported for another forest site in the Amazon (Rizzo et
al., 2013). The observed changes in particle optical properties can be explained by the
presence of Saharan dust particulate matter and biomass burning from the Sahel region.
Beside the appropriate transport direction, during the second event the wind circulation
speed was stronger than during the first event. This enhanced the Saharan dust advec-
tion toward ATTO and resulted in more substantial effects on particle chemical compo-
sition and optical properties at the site.

The third event that began around 10 April and lasted until 17 April, according

to the African AERONET sites (Figure 4.a), and was the largest dust outbreak event
that occurred during the campaign. This was corroborated by the significant increase
registered in the AERONET AOD and by the large values and spread of AOD retrieved
by MODIS (Figure 5.d). However, this massive dust transport toward the Atlantic basin
did not translate into significant changes in the properties of particles sampled at the



ATTO site. As happened in the first dust outbreak event, atmospheric wind circulation
prevented the transport of the dust plume in the direction of the ATTO site. At this time,
a strong zonal wind (eastward) from the African west coast carried the dust plume core
toward the extreme north portion of South America, away from ATTO. Meanwhile, the
site received an influx of air mass predominantly from an area southward of the plume.

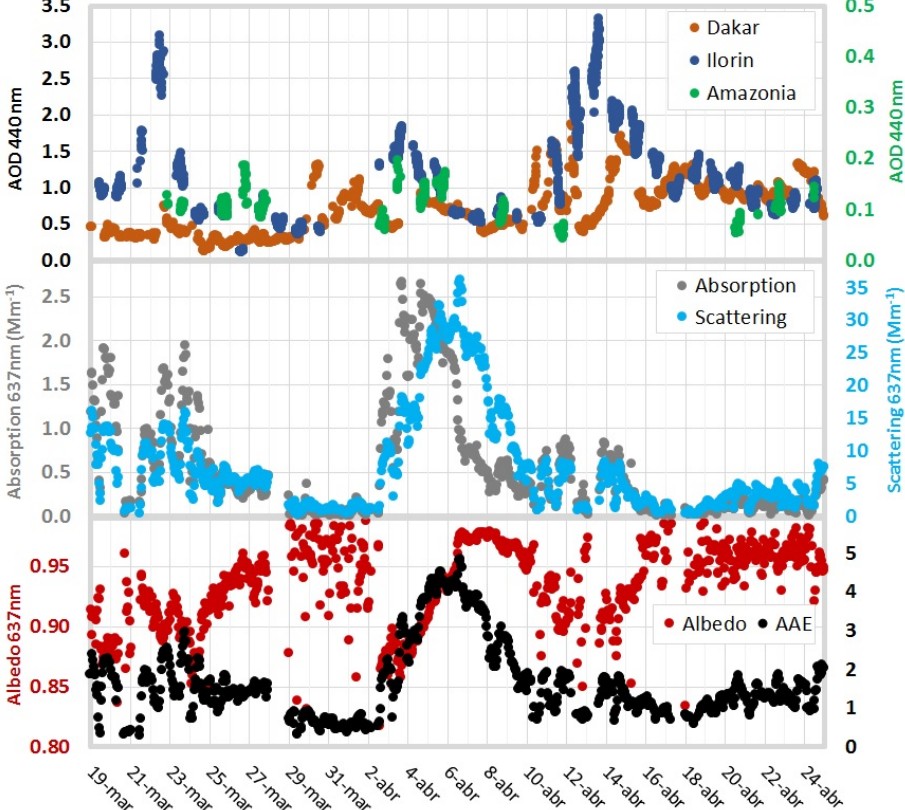


Figure 4. a) Instantaneous Aerosol Optical Depth (AOD) at 440 nm measured at three
AERONET sites: Dakar and Ilorin in Africa, and Embrapa/Manaus in Amazon. b) Par-
ticle absorption and scattering coefficients at 637 nm observed in situ at the ATTO site.
c) Particle single scattering albedo at 637 nm, and Absorption Angstrom Exponent
(AAE), retrieved from in situ observations of aerosol optical properties at ATTO.





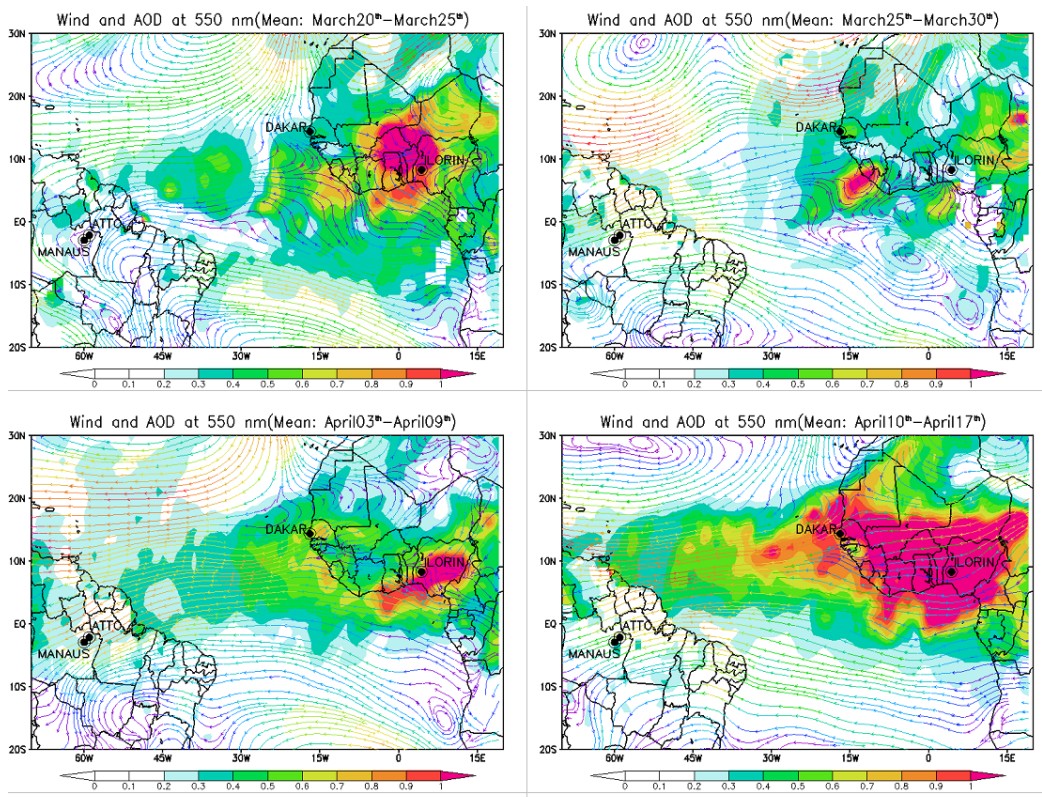

Figure 5. Mean distribution of aerosol optical depth at 550 nm (AOD) and wind at 850

hPa for four distinct periods within the campaign at ATTO site, during the dominance

of: a) the first Saharan dust outbreak; b) a less active dust outbreak period; c) the second

Saharan dust outbreak; d) the third Saharan dust outbreak.

Figure 6 shows that during days without rainfall, the vertical wind speed, W,

was highest above the canopy level (81.65 m), due to the canopy heating the air above

it. Without sunlight forcing, W values did not show significant difference at or below

the canopy (46 m and 36 m height, respectively). Levels below the canopy were, on

average, between a maximum of -0.0012 m/s and minimum of -0.03 m/s, and always

negative, although very close to zero.

At the highest level of W, being positive (ascending air), we also observed the

largest values for Fe(III) and MC.





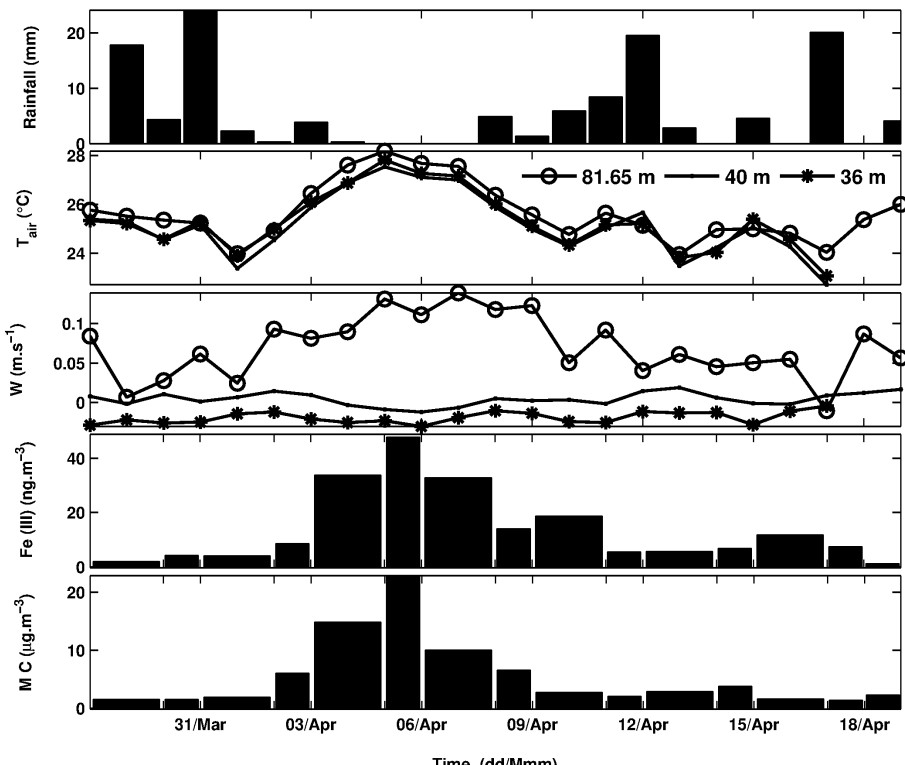

Figure 6. Daily comparison of micrometeorological variables ($T_{air}$, PRP and W) with

measurements of Fe(III) and MC below the canopy.

### 3.4 Spore sample

Sporewatch sample analysis showed that very few coarse particles (> 2 μm di-

ameter) occurred in the atmosphere until 2 April. On 3 April at 13 LT, coarse particles

(2 to 10 μm) peaked in number and were black, hyaline or variously colored and of ir-

regular shape. The amorphous particles were interspersed with a large diversity of small

fungal particles. Fungi that were identified included basidiospores, ascospores,

*Cladosporium, Ganoderma*, and uni- and bi-cellular hyaline conidia (Figure 7). All fun-

gi detected had a diameter less than 12 μm, similar to adjacent coarse dust particles. The

total fungal count was 1,587 spores per cubic meter of air, averaged over 24 h (2-3



April). High concentrations of fungi and other coarse particles persisted, peaking again
at approximately 16:30 LT on 5 April. From the afternoon of 6 April on, very few par-
ticles and only the occasional spore were observed.

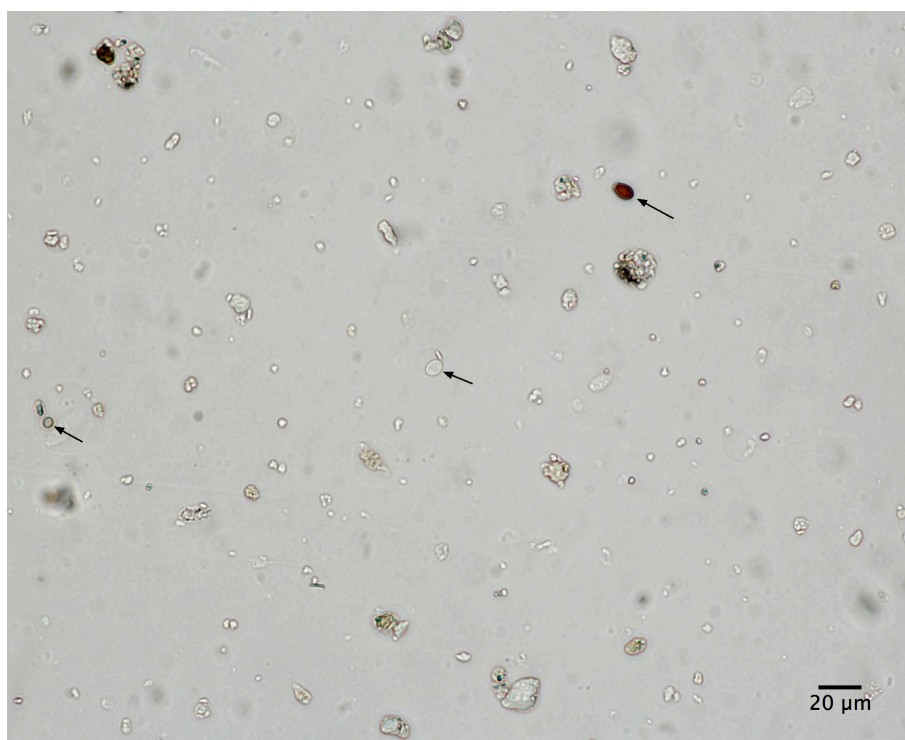


Figure 7. Brightfield microscopy of particles collected in an air sampler at 80 m on 2
April 2015. The arrows point to three fungi: the one on the right is a basidiospore, at
center is a yeast-like conidia, and at left is a small fungal spore of unknown type.

As previously discussed, coarse particles observed between 2 and 6 April are
likely to be associated with the dust cloud arriving from Africa. The large diversity of
small fungal spores and conidia entrained in the dust collected at 80 m height are also
likely to be sourced to Africa. Smoke plumes are known to entrain fungi over long dis-
tances (Mims and Mims, 2004). Dust from Lake Chad is rich in bacteria and fungi (Fa-
vet et al., 2013). These fungi would have also contributed to the elements detected in air



samplers. Bacteria are likely to accompany the dust particles, attached to their surface
(Yamaguchi et al., 2012; Prospero et al., 2005), but it is unknown whether any of these
organisms are still viable upon sedimentation across the central Amazon forest.
Up to half of all micronutrients in the canopy are stored in epiphytes (Cardelus
2010). Fungi housed within lichens take advantage of the large surface area provided by
their algal co-host, and are one of the most bio-absorbent organisms evolved for uptake
of minerals and other nutrients from atmospheric gases and particulates, and from both
dry deposition and rainfall. Another type of fungi common within the canopy are yeasts,
such as Saccharomycetes (Elbert et al., 2007; Womack et al., 2016).
During dry weather, as well as during fog and light rain events, dust deposits
onto the canopy and impacts directly onto leaves of vascular plants (e.g., trees and
vines), as well as epiphytic vascular and non-vascular plants, such as bryophytes (e.g.,
lichens, mosses and liverworts). Dust also settles onto ferns, and fungi within the cano-
py. Air samples from 40 m height showed fungi are common in the canopy. The small-
est and most metabolically active fungi detected in the canopy included lichens and
yeasts (Womack et al., 2015).
Up to 25,000 tons of phosphorus has been calculated as being deposited each
year on the Amazon. Meanwhile, a similar amount of phosphorus has been estimated to
be leached from rainforest soils (Yu et al., 2015). While much of the emphasis has been
on soil chemistry and root absorption, water-soluble minerals, as such as P and K, can
also be absorbed by leaves. Minerals, such as Fe, can be absorbed through plant leaves
as well (Fernandez and Brown, 2013). Thus, canopy deposition of Saharan dust is like-
ly to provide soluble iron to plants via their leaves, in addition to having an influence on
epiphytes and surface microorganisms.





### 3.5 Iron availability

The iron measurement results presented here show a predominance of Fe(III) in the samples, while Fe(II), the form which plants can directly absorb, was measurable only in four samples. The interest in determining the concentration of Fe(II) in aerosols, besides being the ionic form absorbed by plants, is related to its much higher solubility than Fe(III) (Zhu 1997). However, the efficiency in absorbing iron varies among species and genotypes, although within plants the main form is Fe(III) (Kerbauy, 2012).

Therefore, plants develop specific strategies for Fe uptake (Hell and Stephan, 2003; Morrissey and Guerinot, 2009) and, added to this, abiotic factors such as pH, redox state, and temperature can influence mineral nutrient speciation and solubility, as can biotic factors. Plant roots also can modify the rhizosphere to affect nutrient availability; when challenged with a specific nutrient deficiency, plants can induce high-affinity transporters and other mechanisms in their roots, to assist in meeting their mineral nutrient requirements (Grusak, 2001).

The pH of the environmental is important for solubility and therefore the availability of iron to microorganisms. More iron is present in solution in acid soils, but Fe is less soluble in neutral or alkaline situations (Isaac, 1997). The majority of Amazon Basin soils are acidic (Schmink and Wood, 1978) and, similar to Fe, Zn is also better absorbed in soils with low pH (Broadley et al., 2007). In contrast, in alkaline soils, the availability of Zn, Fe and Cu is very low (Marschner, 2012). However, the efficacy of African dust as a fertilizer depends on many factors, such as particulate matter concentration, composition, solubility, and bioavailability of element minerals. In addition, fungi, the most common type of microorganism in the forest (Fracetto et al., 2013), can readily absorb iron, in soluble and insoluble chemical states. Therefore, it is possible



that a small amount of atmospheric iron could affect the microbiota in the canopy, ra-
ther than have a significant effect on soil and root uptake for plants.

Iron availability in the canopy of forests has commonly been found to be limited

for growth of epiphytes, bacteria, and fungi (Crichton, 2009). Addition of iron can have
a variety of effects on plants and fungi, e.g., yeasts grown in iron-limiting culture show
a change in metabolism from fermentation to respiration upon the addition of iron
(Philpott et al., 2012). The ongoing deposition of micronutrients, such as iron, onto the
Amazon biota is likely to increase both epiphytic growth and fungal and bacterial de-
composition within the canopy. Previous observations described an increased tree fall
rate attributed to an abundance of epiphytes (Swap et al., 1992). Increase in iron bioa-
vailability is also known to increase the wood to root ratio, increase the rate of plant
growth, and increase nutrient cycling within a forest (Benzing, 1998; Crichton, 2009;
Cardelius, 2010). The full extent of the influence of Saharan dust is yet to be deter-
mined, although the majority of mineral nutrients available in the soil originate from the
gradual weathering of bedrock in the Amazon basin (Abouchami et al., 2013).

**4 Conclusion**

The current deposition of Saharan dust onto the Amazon is providing an iron-

rich source of essential macronutrients and micronutrients. The atmospheric deposition
of this nutrient-rich dust on the canopy is likely have an influence on rainforest ecology.
Previously unconsidered changes are likely occurring in growth patterns and decompo-
sition rates within the canopy, which affect carbon storage, release, and cycling in the
Amazon.

Overall, this study examined the bioavailability of soluble macro and micronu-

trients to plants of the Amazon Basin, and reported peaks in soluble Fe(III), Fe(II), Na,



Ca, K, and Mg during a major dust transport event from the Saharan desert, according
to meteorological (backward trajectories and wind field), remote sensing (aerosol opti-
cal depth), and *in situ* data analysis. In this way, the elemental contents of samples were
correlated with the arrival of African aerosols.
Our study also reported on the amount of soluble iron in two oxidation states,
Fe(II) and Fe(III), to understand how much of this element is bioavailable to the rain-
forest in the wet season.
Because these nutrients are added to the Amazon by atmospheric deposition they
will likely: 1) directly affect fungi within the canopy, as well as fungal-associated epi-
phytes, such as lichens. 2) have an influence on bacteria, and 3) provide nutrients direct-
ly to leaves and roots of other plants.

**Author contribution**

All authors contributed to the work presented in this paper. R.H.M. Godoi,
C.G.G. Barbosa, J.A. Rizzolo, A.F.F. Godoi, C. Pöhlker and A.O. Manzi developed the
concept, designed the study and the experiments and J. Rizzolo and I.H. Angelis carried
them out. C.I. Yamamoto, G. Borillo and A.O. Manzi provided reagents and gave ana-
lytical-technical support. C. Pöhlker, J. Saturno, D. Moran-Zuloagal and M.O. Sá col-
lected and analyzed data. R.H.M. Godoi, C.G.G. Barbosa, J.A. Rizzolo, P.E. Taylor,
L.V. Rizzo, N.E. Rosário, L.V. Rizzo, R.A.F. Souza, R.V. Andreoli, J. Saturno, D. Mo-
ran-Zuloaga and T. Pauliquevis analyzed data. C. Pöhlker, M.O. Andreae, F. Ditas, L.V.
Rizzo, E.G. Alves, T. Pauliquevis and P.E. Taylor gave conceptual advice. J.A. Rizzolo
prepared the manuscript and, with contributions from C.G.G. Barbosa, A.F.L. Godoi,
E.G. Alves, C. Pöhlker, C.I. Yamamoto, J. Saturno, D. Moran-Zuloaga, L.V. Rizzo,
N.E. Rosário, T. Pauliquevis, M.O. Andreae, P.E. Taylor and R.H.M. Godoi, discussed
the results and implications at all stages.








**Acknowledgments**

We acknowledge the support of the Fundação de Amparo à Pesquisa do Estado

do Amazonas (FAPEAM) and the Financiadora de Estudos e Projetos (FINEP). We
acknowledge logistical support from the Central Office of the Large Scale Biosphere
Atmosphere Experiment in Amazonia (LBA), the Instituto Nacional de Pesquisas da
Amazônia (INPA) and the Universidade do Estado do Amazonas (UEA). We also thank
the Max Planck Society and INPA for continuous support. We acknowledge the support
by the German Federal Ministry of Education and Research (BMBFcontract
01LB1001A) and the Brazilian Ministério da Ciência, Tecnologia e Inovação
(MCTI/FINEP contract 01.11.01248.00) as well as the UEA, FAPEAM, LBA/ INPA
and SDS/CEUC/RDS-Uatumã. We would like to especially thank all the people in-
volved in the logistical support of the ATTO project, in particular Reiner Ditz and Her-
mes Braga Xavier. We acknowledge the micrometeorological group of the INPA/LBA
for their collaboration concerning the meteorological parameters, with special thanks to
Marta Sa and Antonio Huxley. The authors would like to thank Dr. Jose Henrique Pe-
reira from Lawrence Berkeley National Laboratory for the enthusiastic and helpful sug-
gestions.

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
