# Peer review of "Soluble iron nutrients in Saharan dust over the central Amazon rainforest"

_Atmospheric Chemistry and Physics, 2016_

## Referee Comment (RC1) · Anonymous Referee #1 · 8 Sep 2016

General Comments:

Rizzolo et al, address an actual topic in their study which is to quantify and assess the potential impact of mineral dust on the biogeochemistry of the Amazonian basin. Although this is a very important aspect as they correctly present it, the main topic Fe (II)/Fe (III) availability is not well described. In some aspect they are too brief such that their final results cannot be appreciated. However, it is an important topic that meets the requirements of publication at ACP. I would recommend that major revisions are done prior to consideration of a possible publication.

There is not enough evidence that the high soluble content they observe is due to Saharan dust. They need to show some correlations which lead to such arguments.

[Figure]

Simply because the air mass originates from Africa does not imply all chemical components in the aerosol are from the Saharan Desert. This aspect has to be clarified and the identification of the sources that could lead to different soluble iron concentrations in this region has to be discussed. There should be more clarity on the experimental approach on how they measured the soluble metals.

Specific Comments:

How long did the samples stay in the nitric acid flask and were they any measurements done to ensure that the transition from Fe (II) to Fe (III) or vice versa did not occur during the time frame of storage?

Where the extracts filtered? The experimental part is too brief to follow the obtained results.

What were the concentrations of the total fractions of each of the elements? How could the fungi be optimally identified using a light microscope? Was there an algorithm that matched the shapes of the fungi to given types of fungi in a library or was it simply done by intuition?

How good could the mineral dust adsorption in the black carbon signal be isolated since it's mentioned that mineral dust could also produce similar adsorption signals?

The soluble Fe concentration seems to be high for pure Saharan dust. It is known that in lower pH the solubility can be as high as 10% or more. However, it would be helpful to explain in detail how the samples were prepared.

How did you differentiate the sources of the soluble Fe? As mention in the literature comparisons, Fe has other sources, such as combustion, industrial emissions. Iron from these sources in some cases has been found to be more soluble than in Saharan dust. It is unclear from your arguments, why you allocate the soluble iron to be originating from Saharan dust. Long range transport is a mixture of different air masses and these air masses may have different chemical compositions. Thus a more detailed

tracer analysis or correlation of an intrinsic dust element such as Al or Ti would be helpful to identify how significant the Saharan dust contributed to the obtained soluble iron concentrations.

What could be the likely reason for the high wind and high Fe (III) correlation observed above the canopy?

How good did the soluble content correlate with the BC content and total mass concentration?
* * *

---

## Referee Comment (RC2) · Anonymous Referee #2 · 22 Sep 2016

Atmos. Chem. Phys. Discuss., doi:10.5194/acp-2016-557, 2016 MS: Mineral nutrients in Saharan dust and their potential impact on Amazon rain forest ecology Authors: Rizzolo et al.

Abstract

1. the problem of this manuscript that begins by the Title and the Abstract is the generalization of ideas. First I suggest that the authors be more specific in the Title, it should focus on the soluble fraction Fe(II)/Fe(III) issue, which is something important but not enough to account a full story about the Amazon rainforest ecology; 2. In the introduction, clarify at Line 124(pag 6) when the authors say "...Considering that iron is absorbed by plants only as soluble Fe(II)/Fe(III)", previously the authors have stated

that Fe(III) was also recovered by the action of rhizosphere; 3. In methods, Line 142, pag 6, provide filter porosity; 4. In Item 2.5, specify how the samples were storage and if the observations were conducted in the filed or in a particular laboratory condition? 5. The sentence in Line 229-231 "The mass concentration of particles over the Amazon Basin in the wet season is typically around 10 $\mu$g m-3 in locations that are influenced by biomass burning emissions", . . . here there is confusion on the wet season and biomass burning season. Which reference is attributed to the mass concentration mentioned? 6. Part of the text in Lines 254 to 268 (pags 11-12) could be placed at Material and Methods. If possible the authors could place BCe time series superimposed to mass concentrations at Figure 1. This could give some idea on the contribution of BC to the bulk atmospheric concentrations, or if they are lagged in time; 7. In Table 2, how the elemental analysis was conducted for Cu, Zn, Na, Ca, K and Mg ?? and about the NH4 ? 8. In the title of Table 2, it is not "aerosol characterization" it is aerosol composition; it does not correspond to "during the Saharan dust event ", it is before, along and after the event; 9. In Lines 279-282 the authors say that K, Zn and Cu are of biogenic sources, probably mostly emitted during biomass burning. If the detected pulse of dust in this work is coincident with an African biomass burning event as pointed by the authors, what is the level of certainty to say that their main source is the mineral fraction? 10. In Line 322, the comparison of the present work with Andreae et al. (2015): does both work have same methods and associated errors? Results of Andreae et al. (2015) correspond to what period of the year. Specify please. 11. Text in lines 333-338 is unnecessary; 12. Dates in Figure 2 is unreadable; 13. Figure 3 should be completely edited. It is not possible to use the Hysplit output directly; 14. For Figure 3, use ensembles, not a single trajectory; a family of trajectories gives a better idea of all geographical contributions; 15. Lines 369-374; Figure 1 shows before, along and after the "dust storm", I suggest that the authors run the Hysplit model in these 3 circumstances and then make their conclusions; 16. In Line 387 provide complete localization of the three AERONET sites: Dakar and Ilorin in Africa, and Embrapa/Manaus in Amazon; 17. In Figure 5, AOD do not distinguish dust from biomass

burning products. From the location of higher AODs in the diagrams it seems that your source could have some contribution from biomass burning than mineral dust. Also the results presented in the Hysplit are not totally in accordance to wind flows at the charts at Figure 5. Maybe the source is a net combination of both; I strongly suggest that the authors add a map with fire spots for the period of sampling, so to make better differentiate; 18. In Figure 6, what is MC ? Please, correct the legend of time; 19. The discussion on fungi is very poor. There is none description of the species nor anything on their biogeography. The lesson of this result is the fact that a more detail aerobiological research should be conducted to be published; 20. In Line 463-465, the authors say "Smoke plumes are known to entrain fungi over long distances (Mims and Mims, 2004). Dust from Lake Chad is rich in bacteria and fungi." Here becomes explicitly that the authors are not able to stablish a source of the particulate matter entering Amazon in the considered event: Saharan mineral dust or sub-Saharan biomass burning ?? 21. The Amazon itself is a fantastic source of bacteria and fungi, and only an endemic specie of Africa, detected in Amazon, at high level (ex. The top of the ATTO) could make a clear distinction; 22. In item 3.5 the authors says that "a small amount of atmospheric iron could affect the microbiota in the canopy, rather than have a significant effect on soil and root uptake for plants." This is an speculation and from this work it is not possible to conclude anything; 23. In my opinion, most of item 3.5 is Introduction to the study since most of the text is compilation from the literature associated to this work. 24. The conclusion unrealistic, should be reduced to the basic findings.

GENERAL OPINNION: The positive issue in this work is the measurement of Fe(II) in dust, this is great and I encourage to be improved in the future. From Table 2, it is clear that the event provided only 4 measurements, and 2 of them are not statistically significant. Most important is that they occurred during the storm. The tentative of describing the transport mechanism and sources (desert and/or) biomass burning needs improvements. There are many text here that only explain the general sense of the problem and definitively the fungi issue should be excluded. Maybe focused in a specific work or using more data. Most important to have in mind is that between the

mineral apportionment and the ecological response (in all levels) there is a long way of processes that this work do not account alone. From the above, I recommend this work as a letter or communication not as a Research paper.

---

## Author Comment (AC1) · 14 Jan 2017

General Comments: Although this is a very important aspect as they correctly present it, the main topic Fe (II)/Fe (III) availability is not well described. In some aspect they are too brief such that their final results cannot be appreciated. However, it is an important topic that meets the requirements of publication at ACP. I would recommend that major revisions are done prior to consideration of a possible publication.

1. There is not enough evidence that the high soluble content they observe is due to Saharan dust. They need to show some correlations which lead to such arguments. R: Yes, we agree with the reviewer's comment. The text was adjusted to show the correlations of the soluble contents with physical and chemical properties of long-range

transported aerosol plumes. We are confident that the manuscript in its revised version convincingly shows that indeed a pronounced Saharan dust plume has been collected, which is the only plausible explanation for the increased Fe concentrations. Our argumentation is further supported by a broad set of related literature that is consistent with our observations. In general, this experiment was designed to obtain soluble iron measurements in aerosols originating from Saharan dust plumes. To achieve this goal, the following strategies were adopted: Location - the progress of African dust occurrences across the Amazon can be readily followed by satellite images, which typically show that it takes about 1 week for dust outbreaks to cross to the Atlantic (Yu et al., 2015). The sampling site is located in the common transport pathway of Saharan dust and receives dust laden air masses on seasonal cycles. Period of the year - several papers have shown that Saharan dust reaches the Amazon Basin especially during the months of March/April/May: (L.106-108) "The sampling period ranged from 19 March to 25 April 2015, which is within the typical season when dust transport to the Amazon Basin has been reported before (Talbot et al., 1990; Swap et al., 1992; Prospero et al., 2014; Yu et al., 2015)" Iron source - studies have shown that in Central Amazonia, the iron content in aerosols originates from crustal sources or long range transport: (L. 349-351): "Soil dust related elements are typically present at the highest concentrations during the early wet-to-dry season transition (May), as has been shown in previous studies (Pauliquevis et al., 2012; Andreae et al., 2015)" (L. 364-367): "Pauliquevis et al. (2012) also observed increases in the concentration of total Fe with values reaching 60 ng m $\dot{I}\tilde{u}$ $^3$ in the fine mode mostly during February to April in the Amazon Basin, with a seasonal average of 36 ng m $\dot{I}\tilde{u}$ $^3$. They attributed this to episodes of Saharan dust transport."

2. Simply because the air mass originates from Africa does not imply all chemical components in the aerosol are from the Saharan Desert. This aspect has to be clarified and the identification of the sources that could lead to different soluble iron concentrations in this region has to be discussed. R: We agree with the reviewer. The long-range transported aerosol from Africa typically comprises a complex mixture of desert dust,

biomass burning smoke, maritime aerosols, and biological particles. The revised discussion on the elements measured in the aerosol samples and their potential origin from the aforementioned sources has been extended. We show that the dust fraction is the only plausible explanation for most of the Fe content, whereas Na, Cl, Mg, K, and other elements are associated with sea spray and biomass burning aerosol fractions in the long-range transport plumes. Specifically, we are referring to the Sahara and Sahel region together as "Sahara". This has been stated in the abstract. In addition to mineral dust aerosol from the Sahara and/or Sahel region, there is also marine aerosol introduced from the Atlantic Ocean, which can contribute sea salt elements and biogenic sulfate. Because the northern hemisphere winter is also the fire season in West Africa, biomass burning smoke often arrives together with African dust. This can be seen in the BCe concentration of up to 0.3 $\mu$g m Ìű $^3$ on 4 April. At a typical fraction of 7% of BC in biomass TPM, this could contribute about 4 $\mu$g m Ìű $^3$ of TPM, or about 20% of the TPM at the peak of the BCe concentration. We have added a comment on this in the text. But regarding our main goal, Longo et al. (2016) indicated that the iron in Saharan dust exhibits different oxidation states across the different sampling sites, suggesting that the longer an aerosol remains in the atmosphere, the more reduced the iron becomes. These complex interactions suggest that the various particle-aging mechanisms, such as acidic reactions and photoreduction, may be working simultaneously. (L. 221-224) Although the Hysplit backward trajectories do not guarantee that pure end members were sampled, they help to demonstrate that most air masses were of North African origin during the sampling time periods. (L. 357-361) The concentration of iron (II) obtained recently at Barbados is in the same order of magnitude as the amounts that we measured (Zhu et al., 1997). Additionally, the increase in particle number measured during the sampling period correlated with the presence of Saharan aerosols at the ATTO site.

3. There should be more clarity on the experimental approach on how they measured the soluble metals. R: We agree with the reviewer. The experimental details in section 2.2 have been specified in more detail (L.128-146) "All the analyses were performed

on the TPM soluble fraction. Each sampled filter immersed in nitric acid solution was extracted by an ultrasonic bath for 10 mins. The extract of each sample was filtered with a polyvinylidene difluoride (PVDF) sterile membrane (0.22 $\mu$m pore size, diameter 25 mm, Millipore, Merck) and analysed by ion chromatography (ICS 5000, Dionex-Thermo Scientific, USA). For the transition metal quantification and iron speciation, pyridine-2,6-dicarboxylic acid (PDCA) was used as eluent and 4-2-2-pyridyl resorcinol (PAR) was used as a post-column reagent, stabilized by a PC-10 nitrogen pump. The system flow was 0.3 mL min Ì ű 1 through an IonPac CG5A (2 x 50 mm) guard column, CS5A capillary column (2 x 250 mm) and UV-Vis spectrophotometry with detection at 530 nm (Cardellicchio et al., 1997). For soluble Fe (II), Fe (III), Cu and Zn the detection limits (USEPA, 1997) were 1.7, 0.4, 1.3, 4.1 $\mu$g L Ì ű 1, respectively and the expanded uncertainties at the 95% level of confidence (BIPM, 2008) were of 3, 42, 46, 56 %, respectively. For the cation analysis, ultrapure water and methanesulfonic acid (MSA) was used as the eluent at a 20 mM constant concentration, with automatic suppression (CSRS suppressor - 2 mm), and with a 0.33 mL min Ì ű 1 system flow through an IonPac CG-12 guard column (2 x 50 mm) and CS-12 (2 x 250 mm) capillary column. This resulted in a 14 min running time for each injection. For soluble Na, $NH_4+$, K, Mg and Ca the detection limits (USEPA, 1997) were 2.0, 1.3, 0.9, 0.7, 1.8 $\mu$g L Ì ű 1, respectively, and the expanded uncertainties at the 95% level of confidence (BIPM, 2008) were of 9, 7, 21, 11, 23 %, respectively."

Specific Comments:

4. How long did the samples stay in the nitric acid flask and were they any measurements done to ensure that the transition from Fe (II) to Fe (III) or vice versa did not occur during the time frame of storage? R: The samples stayed in the nitric acid flask during the sampling period of March 19th to April 25th and no test was performed during the storage period. Based on Cwiertny et al (2008), the dissolved Fe(II) / Total dissolved Fe ratio of Saharan Dust, is practically constant over time: "Nitric acid suppresses the formation of iron(II) at low pH; therefore, pH can also act as a control of oxidation state

of aerosol iron."

5. Where the extracts filtered? The experimental part is too brief to follow the obtained results. R: The method section was rewritten with more details and the required information was added to the text as follows: (L. 128-132) "Each sampled filter immersed in nitric acid solution was extracted by an ultrasonic bath for 10 mins. The extract of each sample was filtered through a polyvinylidene difluoride (PVDF) sterile membrane, 0.22 $\mu$m pore size, diameter 25 mm (Millipore, Merck) and analysed by ion chromatography (ICS 5000, Dionex-Thermo Scientific, USA)."

6. What were the concentrations of the total fractions of each of the elements? R: The focus of the sampling efforts was to quantify the soluble fraction. In order to keep the oxidation state of iron stable, we immersed the filters in the acidic solution. The chemical analysis performed was Ion Chromatography and to assess the total fraction we used TPM elemental composition results sampled simultaneously at another site, as detailed in Table 1: (L.269-272). Previous studies have shown that the aerosol composition and burden is almost identical at the two sites when long-range transport is dominant, as it was during our study.

7. How could the fungi be optimally identified using a light microscope? Was there an algorithm that matched the shapes of the fungi to given types of fungi in a library or was it simply done by intuition? R: The fungi types were identified using consolidated data bases and a certified spore counter with the US National Allergy Bureau: (L. 214-215).

8. How good could the mineral dust adsorption in the black carbon signal be isolated since it's mentioned that mineral dust could also produce similar adsorption signals? R: The absorption Angstrom exponent (AAE) can be used to investigate the relative contribution of different particle sources to the BCe signal. Particle samples impacted by mineral dust typically show AAE greater than 2.0, while soot from fossil fuel combustion shows AEE close to 1.0 (e.g., Bergstrom et al., 2007). There are methods proposed in the literature to distinguish between fossil fuel soot and other light absorbing particles (e.g., Lack and Langridge, 2013), but we consider that this is not in the scope of the present article. However, to clarify this point, we reformulated part of sections 2.4 and 3.1 as follows: (L. 164-174): "Soot, mineral dust, and biogenic particles are light absorbers (Moosmüller et al., 2009; 2011; Guyon et al., 2004; Andreae and Gelencsér 2006) and may contribute to the observed BCe signal. The relative contributions of particle sources to BCe can be investigated by considering the absorption spectral variability, by means of the so called Absorption Ångström Exponent (AAE). Soot from fossil fuel combustion typically shows AAE close to 1.0, while particles impacted by dust emissions show AAE greater than 2 (Bergstrom et al., 2007). Studies indicate that samples impacted by biomass burning aerosols show AAE in the range of 1.5-2.0 (Bergstrom et al., 2007; Rizzo et al., 2011). The spectral dependency of particle absorption coefficients was monitored using a 7-wavelength Aethalometer (Model AE33, Magee Scientific Company, USA, $\lambda$ = 370, 470, 520, 590, 660, 880, and 950 nm), compensated for filter loading and multiple scattering effects (Rizzo et al., 2011)." (L. 237-253): "Particulate Fe(III) and Fe(II) concentrations increased between 3 and 9 April, simultaneously with an increase in particle absorption and scattering coefficients (Figure 4.a). A decrease was observed in the intrinsic property, single scattering albedo (SSA, Figure 4.b), suggesting the presence of particles that are efficient light absorbers, such as soot from fossil fuel combustion and biomass burning, mineral dust, and biogenic particles. For comparison, Rizzo et al. (2013) reported that a 7% decrease in SSA at another forest site in the central Amazon during the wet season periods proved to be related to advection of African aerosols. The spectral dependency of absorption, AAE, can be used to distinguish between the different sources of light absorbing particles. The elevated AAE values observed between 6 and 10 April (Figure 4.b) contradict the influence of soot from fossil fuel combustion. During the clean periods (25 March to 2 April and 16 to 24 April), dominated by biogenic particles, AAE values were around 1.8, so that this source of particles, ever present at Amazonian forest sites, may not have contributed to the AAE increase between 6 and 10 April. Therefore, two light absorbing particle sources are left to explain the increase in absorption and AAE values: biomass burning and mineral dust particles. Fire activity is typically low in the central Amazon between November and April, with less than 2 fire spots per 1000 km2 and day on average (Castro-Videla et al., 2013), which is corroborated by the map of fire spots distribution during the campaign period (Figure 5)." (L. 286-289): "Our conclusion that African dust dominates the aerosol budget during the dust event is in agreement with Castro Videla et al. (2013), who, based on a five-year study, concluded that peaks in AOD in the central Amazon during the wet season had a significant contribution from coarse mode particles, pointing to a major role of African advection."

9. The soluble Fe concentration seems to be high for pure Saharan dust. It is known that in lower pH the solubility can be as high as 10% or more. However, it would be helpful to explain in detail how the samples were prepared. R: Yes, we agree with the reviewer and the methods section was modified: (L. 128-146). Regarding the required information, each sampled filter immersed in nitric acid solution was extracted by ultrasonic bath. The extract of each sample was filtered through a PVDF sterile membrane, 0.22 $\mu$m pore size, diameter of 25 mm and analyzed by ion chromatography for NaâĄž, NHâĆĎâĄž, KâĄž, Mg$^2$âĄž, Ca$^2$âĄž, Fe3+, Cu2+, Zn2+ and Fe2+.

10. How did you differentiate the sources of the soluble Fe? As mention in the literature comparisons, Fe has other sources, such as combustion, industrial emissions. Iron from these sources in some cases has been found to be more soluble than in Saharan dust. It is unclear from your arguments, why you allocate the soluble iron to be originating from Saharan dust. Long range transport is a mixture of different air masses and these air masses may have different chemical compositions. Thus a more detailed tracer analysis or correlation of an intrinsic dust element such as Al or Ti would be helpful to identify how significant the Saharan dust contributed to the obtained soluble iron concentrations. R: The reviewer is correct. It is very hard to determine sources based on trajectories/long transport alone. The ATTO site, especially its location, was chosen to provide the most reliable sampling of natural tropical aerosols in the world.

It is in one of most remote areas on the planet, and this is the primary reason why we assert that our aerosol samples were mostly from natural contributions. As explained in the answer to the first question, air masses reaching the site have crossed more than 1500 km of primary rainforests in the Amazon Basin. The bulk of the background aerosol at ATTO is biogenic. There are no industries in the path of these air masses. (L. 273-283): "Biomass burning in Africa could also have contributed some of the observed Fe, but unfortunately, little is known about Fe emissions from savanna fires, and the available data span a wide range. From the work of Gaudichet et al. (1995), one can derive a Fe content of 0.016% in savanna smoke TPM, which, at a peak biomass smoke concentration of 4 $\mu$g m Ìǔ 3, would only give 0.6 ng Fe m Ìǔ 3. Using the BC/Fe ratio of ca. 40 from Maenhaut et al. (1996) and the peak BCe concentration of 0.3 $\mu$g m Ìǔ 3, we can estimate ca. 8 ng Fe m Ìǔ 3. Finally, using the Fe emission factor of 0.026 g kg Ìǔ 1 d.m. for African savanna fires from Andreae et al. (1998) and the BC emission factor of 0.6 g kg Ìǔ 1 from Andreae and Merlet (2001 and updates), we can estimate a peak pyrogenic Fe contribution of 13 ng m-3. This compares to 64 ng m Ìǔ 3 of soluble iron (details follow in section 3.3) at the same time, and, given that only a small fraction of the Fe in biomass smoke is likely to be soluble, it is clear that the dominant fraction of soluble Fe comes from the African mineral dust." We are quite confident of our results because when we compare with dust obtained at Barbados, we observed that the soluble ferrous iron (Fe(II)) and Total soluble iron (Fe) are in the same order of magnitude (Zhu et al., 1997).

11. What could be the likely reason for the high wind and high Fe (III) correlation observed above the canopy? R: These vertical transport correlations could be analyzed by micrometeorological methods (e.g., eddy covariance). However, this analysis is beyond the scope of this work. The idea of presenting the vertical speed was only to show that regardless of the cloudy conditions, the W values are close to zero, indicating that the vertical transport is very weak. Thus, the text was rewritten to make this clear. The vertical wind data have been removed.

12. How good did the soluble content correlate with the BC content and total mass concentration? R: We thank the reviewer for this opportune suggestion, and added the time series of BCe and particle soluble fraction concentration to Figure 2. This clearly shows that the latter are well correlated with total mass concentration, especially during the event from April 1 to 8. We also added the following sentence to the section 3.2: (L. 321-324) "Figure 2 and Table 1 show that BCe concentrations significantly increased regionally during 1-8 April, coinciding with the increase in PM10 and particle soluble fraction concentrations and indicating that some biomass smoke (probably from fires in West Africa) arrived together with the dust".

Please also note the supplement to this comment:
http://www.atmos-chem-phys-discuss.net/acp-2016-557/acp-2016-557-AC1-supplement.pdf

**Supplement:**

*Response to Referee #1*

**General Comments:**
**Although this is a very important aspect as they correctly present it, the main topic Fe (II)/Fe (III) availability is not well described. In some aspect they are too brief such that their final results cannot be appreciated. However, it is an important topic that meets the requirements of publication at ACP. I would recommend that major revisions are done prior to consideration of a possible publication.**

**1. There is not enough evidence that the high soluble content they observe is due to Saharan dust. They need to show some correlations which lead to such arguments.**
*R: Yes, we agree with the reviewer's comment. The text was adjusted to show the correlations of the soluble contents with physical and chemical properties of long-range transported aerosol plumes. We are confident that the manuscript in its revised version convincingly shows that indeed a pronounced Saharan dust plume has been collected, which is the only plausible explanation for the increased Fe concentrations. Our argumentation is further supported by a broad set of related literature that is consistent with our observations. In general, this experiment was designed to obtain soluble iron measurements in aerosols originating from Saharan dust plumes. To achieve this goal, the following strategies were adopted:*

*Location - the progress of African dust occurrences across the Amazon can be readily followed by satellite images, which typically show that it takes about 1 week for dust outbreaks to cross to the Atlantic (Yu et al., 2015). The sampling site is located in the common transport pathway of Saharan dust and receives dust laden air masses on seasonal cycles.*

*Period of the year - several papers have shown that Saharan dust reaches the Amazon Basin especially during the months of March/April/May: (L.106-108) "The sampling period ranged from 19 March to 25 April 2015, which is within the typical season when dust transport to the Amazon Basin has been reported before (Talbot et al., 1990; Swap et al., 1992; Prospero et al., 2014; Yu et al., 2015)"*

*Iron source - studies have shown that in Central Amazonia, the iron content in aerosols originates from crustal sources or long range transport: (L. 349-351): "Soil dust related elements are typically present at the highest concentrations during the early wet-to-dry season transition (May), as has been shown in previous studies (Pauliquevis et al., 2012; Andreae et al., 2015)" (L. 364-367): "Pauliquevis et al. (2012) also observed increases in the concentration of total Fe with values reaching 60 ng $m^{-3}$ in the fine mode mostly during February to April in the Amazon Basin, with a seasonal average of 36 ng $m^{-3}$. They attributed this to episodes of Saharan dust transport."*

**2. Simply because the air mass originates from Africa does not imply all chemical components in the aerosol are from the Saharan Desert. This aspect has to be clarified and the identification of the sources that could lead to different soluble iron concentrations in this region has to be discussed.**
*R: We agree with the reviewer. The long-range transported aerosol from Africa typically comprises a complex mixture of desert dust, biomass burning smoke, maritime aerosols, and biological particles. The revised discussion on the elements measured in the aerosol samples and their potential origin from the aforementioned sources has been extended. We show that the dust fraction is the only plausible explanation for most of the Fe content, whereas Na, Cl, Mg, K, and other elements are associated with sea spray and biomass burning aerosol fractions in the long-range transport plumes.*

*Specifically, we are referring to the Sahara and Sahel region together as "Sahara". This has been stated in the abstract. In addition to mineral dust aerosol from the Sahara and/or Sahel region, there is also marine aerosol introduced from the Atlantic Ocean, which can contribute sea salt elements and biogenic sulfate.*

*Because the northern hemisphere winter is also the fire season in West Africa, biomass burning*

*smoke often arrives together with African dust. This can be seen in the BCe concentration of up to 0.3 µg m⁻³ on 4 April. At a typical fraction of 7% of BC in biomass TPM, this could contribute about 4 µg m⁻³ of TPM, or about 20% of the TPM at the peak of the BCe concentration. We have added a comment on this in the text.*

*But regarding our main goal, Longo et al. (2016) indicated that the iron in Saharan dust exhibits different oxidation states across the different sampling sites, suggesting that the longer an aerosol remains in the atmosphere, the more reduced the iron becomes. These complex interactions suggest that the various particle-aging mechanisms, such as acidic reactions and photoreduction, may be working simultaneously.*

*(L. 221-224) Although the Hysplit backward trajectories do not guarantee that pure end members were sampled, they help to demonstrate that most air masses were of North African origin during the sampling time periods.*

*(L. 357-361) The concentration of iron (II) obtained recently at Barbados is in the same order of magnitude as the amounts that we measured (Zhu et al., 1997). Additionally, the increase in particle number measured during the sampling period correlated with the presence of Saharan aerosols at the ATTO site.*

**3. There should be more clarity on the experimental approach on how they measured the soluble metals.**

*R: We agree with the reviewer. The experimental details in section 2.2 have been specified in more detail (L.128-146) "All the analyses were performed on the TPM soluble fraction. Each sampled filter immersed in nitric acid solution was extracted by an ultrasonic bath for 10 mins. The extract of each sample was filtered with a polyvinylidene difluoride (PVDF) sterile membrane (0.22 µm pore size, diameter 25 mm, Millipore, Merck) and analysed by ion chromatography (ICS 5000, Dionex-Thermo Scientific, USA).*

*For the transition metal quantification and iron speciation, pyridine-2,6-dicarboxylic acid (PDCA) was used as eluent and 4-2-2-pyridyl resorcinol (PAR) was used as a post-column reagent, stabilized by a PC-10 nitrogen pump. The system flow was 0.3 mL min⁻¹ through an IonPac CG5A (2 x 50 mm) guard column, CS5A capillary column (2 x 250 mm) and UV-Vis spectrophotometry with detection at 530 nm (Cardellicchio et al., 1997). For soluble Fe (II), Fe (III), Cu and Zn the detection limits (USEPA, 1997) were 1.7, 0.4, 1.3, 4.1 µg L⁻¹, respectively and the expanded uncertainties at the 95% level of confidence (BIPM, 2008) were of 3, 42, 46, 56 %, respectively.*

*For the cation analysis, ultrapure water and methanesulfonic acid (MSA) was used as the eluent at a 20 mM constant concentration, with automatic suppression (CSRS suppressor - 2 mm), and with a 0.33 mL min⁻¹ system flow through an IonPac CG-12 guard column (2 x 50 mm) and CS-12 (2 x 250 mm) capillary column. This resulted in a 14 min running time for each injection. For soluble Na, $NH_4^+$, K, Mg and Ca the detection limits (USEPA, 1997) were 2.0, 1.3, 0.9, 0.7, 1.8 µg L⁻¹, respectively, and the expanded uncertainties at the 95% level of confidence (BIPM, 2008) were of 9, 7, 21, 11, 23 %, respectively."*

**Specific Comments:**

**4. How long did the samples stay in the nitric acid flask and were they any measurements done to ensure that the transition from Fe (II) to Fe (III) or vice versa did not occur during the time frame of storage?**

*R: The samples stayed in the nitric acid flask during the sampling period of March 19ᵗʰ to April 25ᵗʰ and no test was performed during the storage period. Based on Cwiertny et al (2008), the dissolved Fe(II) / Total dissolved Fe ratio of Saharan Dust, is practically constant over time: "Nitric acid suppresses the formation of iron(II) at low pH; therefore, pH can also act as a control of oxidation state of aerosol iron."*

**5. Where the extracts filtered? The experimental part is too brief to follow the obtained results.**

*R: The method section was rewritten with more details and the required information was added to the text as follows: (L. 128-132) "Each sampled filter immersed in nitric acid solution was extracted by an ultrasonic bath for 10 mins. The extract of each sample was filtered through a polyvinylidene difluoride (PVDF) sterile membrane, 0.22 µm pore size, diameter 25 mm (Millipore, Merck) and analysed by ion chromatography (ICS 5000, Dionex-Thermo Scientific, USA)."*

**6. What were the concentrations of the total fractions of each of the elements?**

*R: The focus of the sampling efforts was to quantify the soluble fraction. In order to keep the oxidation state of iron stable, we immersed the filters in the acidic solution. The chemical analysis performed was Ion Chromatography and to assess the total fraction we used TPM elemental composition results sampled simultaneously at another site, as detailed in Table 1: (L.269-272). Previous studies have shown that the aerosol composition and burden is almost identical at the two sites when long-range transport is dominant, as it was during our study.*

**7. How could the fungi be optimally identified using a light microscope? Was there an algorithm that matched the shapes of the fungi to given types of fungi in a library or was it simply done by intuition?**

*R: The fungi types were identified using consolidated data bases and a certified spore counter with the US National Allergy Bureau: (L. 214-215).*

**8. How good could the mineral dust adsorption in the black carbon signal be isolated since it's mentioned that mineral dust could also produce similar adsorption signals?**

*R: The absorption Angstrom exponent (AAE) can be used to investigate the relative contribution of different particle sources to the BCe signal. Particle samples impacted by mineral dust typically show AAE greater than 2.0, while soot from fossil fuel combustion shows AEE close to 1.0 (e.g., Bergstrom et al., 2007). There are methods proposed in the literature to distinguish between fossil fuel soot and other light absorbing particles (e.g., Lack and Langridge, 2013), but we consider that this is not in the scope of the present article. However, to clarify this point, we reformulated part of sections 2.4 and 3.1 as follows:*

*(L. 164-174): "Soot, mineral dust, and biogenic particles are light absorbers (Moosmüller et al., 2009; 2011; Guyon et al., 2004; Andreae and Gelencsér 2006) and may contribute to the observed BCe signal. The relative contributions of particle sources to BCe can be investigated by considering the absorption spectral variability, by means of the so called Absorption Ångström Exponent (AAE). Soot from fossil fuel combustion typically shows AAE close to 1.0, while particles impacted by dust emissions show AAE greater than 2 (Bergstrom et al., 2007). Studies indicate that samples impacted by biomass burning aerosols show AAE in the range of 1.5-2.0 (Bergstrom et al., 2007; Rizzo et al., 2011). The spectral dependency of particle absorption coefficients was monitored using a 7-wavelength Aethalometer (Model AE33, Magee Scientific Company, USA, λ = 370, 470, 520, 590, 660, 880, and 950 nm), compensated for filter loading and multiple scattering effects (Rizzo et al., 2011)."*

*(L. 237-253): "Particulate Fe(III) and Fe(II) concentrations increased between 3 and 9 April, simultaneously with an increase in particle absorption and scattering coefficients (Figure 4.a). A decrease was observed in the intrinsic property, single scattering albedo (SSA, Figure 4.b), suggesting the presence of particles that are efficient light absorbers, such as soot from fossil fuel combustion and biomass burning, mineral dust, and biogenic particles. For comparison, Rizzo et al. (2013) reported that a 7% decrease in SSA at another forest site in the central Amazon during the wet season periods proved to be related to advection of African aerosols. The spectral dependency of absorption, AAE, can be used to distinguish between the different sources of light absorbing particles. The elevated AAE values observed between 6 and 10 April*

*(Figure 4.b) contradict the influence of soot from fossil fuel combustion. During the clean periods (25 March to 2 April and 16 to 24 April), dominated by biogenic particles, AAE values were around 1.8, so that this source of particles, ever present at Amazonian forest sites, may not have contributed to the AAE increase between 6 and 10 April.*

*Therefore, two light absorbing particle sources are left to explain the increase in absorption and AAE values: biomass burning and mineral dust particles. Fire activity is typically low in the central Amazon between November and April, with less than 2 fire spots per 1000 km2 and day on average (Castro-Videla et al., 2013), which is corroborated by the map of fire spots distribution during the campaign period (Figure 5).”*

*(L. 286-289): “Our conclusion that African dust dominates the aerosol budget during the dust event is in agreement with Castro Videla et al. (2013), who, based on a five-year study, concluded that peaks in AOD in the central Amazon during the wet season had a significant contribution from coarse mode particles, pointing to a major role of African advection.”*

**9. The soluble Fe concentration seems to be high for pure Saharan dust. It is known that in lower pH the solubility can be as high as 10% or more. However, it would be helpful to explain in detail how the samples were prepared.**

*R: Yes, we agree with the reviewer and the methods section was modified: (L. 128-146). Regarding the required information, each sampled filter immersed in nitric acid solution was extracted by ultrasonic bath. The extract of each sample was filtered through a PVDF sterile membrane, 0.22 μm pore size, diameter of 25 mm and analyzed by ion chromatography for $Na^+$, $NH_4^+$, $K^+$, $Mg^{2+}$, $Ca^{2+}$, $Fe^{3+}$, $Cu^{2+}$, $Zn^{2+}$ and $Fe^{2+}$.*

**10. How did you differentiate the sources of the soluble Fe? As mention in the literature comparisons, Fe has other sources, such as combustion, industrial emissions. Iron from these sources in some cases has been found to be more soluble than in Saharan dust. It is unclear from your arguments, why you allocate the soluble iron to be originating from Saharan dust. Long range transport is a mixture of different air masses and these air masses may have different chemical compositions. Thus a more detailed tracer analysis or correlation of an intrinsic dust element such as Al or Ti would be helpful to identify how significant the Saharan dust contributed to the obtained soluble iron concentrations.**

*R: The reviewer is correct. It is very hard to determine sources based on trajectories/long transport alone. The ATTO site, especially its location, was chosen to provide the most reliable sampling of natural tropical aerosols in the world. It is in one of most remote areas on the planet, and this is the primary reason why we assert that our aerosol samples were mostly from natural contributions. As explained in the answer to the first question, air masses reaching the site have crossed more than 1500 km of primary rainforests in the Amazon Basin. The bulk of the background aerosol at ATTO is biogenic. There are no industries in the path of these air masses.*

*(L. 273-283): “Biomass burning in Africa could also have contributed some of the observed Fe, but unfortunately, little is known about Fe emissions from savanna fires, and the available data span a wide range. From the work of Gaudichet et al. (1995), one can derive a Fe content of 0.016% in savanna smoke TPM, which, at a peak biomass smoke concentration of 4 μg m−3, would only give 0.6 ng Fe m−3. Using the BC/Fe ratio of ca. 40 from Maenhaut et al. (1996) and the peak BCe concentration of 0.3 μg m−3, we can estimate ca. 8 ng Fe m−3. Finally, using the Fe emission factor of 0.026 g kg−1 d.m. for African savanna fires from Andreae et al. (1998) and the BC emission factor of 0.6 g kg−1 from Andreae and Merlet (2001 and updates), we can estimate a peak pyrogenic Fe contribution of 13 ng m-3. This compares to 64 ng m −3 of soluble iron (details follow in section 3.3) at the same time, and, given that only a small fraction of the Fe in biomass smoke is likely to be soluble, it is clear that the dominant fraction of soluble Fe comes from the African mineral dust.”*

*We are quite confident of our results because when we compare with dust obtained at Barbados, we observed that the soluble ferrous iron (Fe(II)) and Total soluble iron (Fe) are in*

*the same order of magnitude (Zhu et al., 1997).*

**11. What could be the likely reason for the high wind and high Fe (III) correlation observed above the canopy?**
*R: These vertical transport correlations could be analyzed by micrometeorological methods (e.g., eddy covariance). However, this analysis is beyond the scope of this work. The idea of presenting the vertical speed was only to show that regardless of the cloudy conditions, the W values are close to zero, indicating that the vertical transport is very weak. Thus, the text was rewritten to make this clear. The vertical wind data have been removed.*

**12. How good did the soluble content correlate with the BC content and total mass concentration?**
*R: We thank the reviewer for this opportune suggestion, and added the time series of BCe and particle soluble fraction concentration to Figure 2. This clearly shows that the latter are well correlated with total mass concentration, especially during the event from April 1 to 8. We also added the following sentence to the section 3.2: (L. 321-324) "Figure 2 and Table 1 show that BCe concentrations significantly increased regionally during 1-8 April, coinciding with the increase in PM10 and particle soluble fraction concentrations and indicating that some biomass smoke (probably from fires in West Africa) arrived together with the dust".*

*References:*

*Andreae, M. O., Andreae, T. W., Annegarn, H., Beer, F., Cachier, H., Elbert, W., Harris, G. W., Maenhaut, W., Salma, I., Swap, R., Wienhold, F. G., and Zenker, T., Airborne studies of aerosol emissions from savanna fires in southern Africa: 2. Aerosol chemical composition: J. Geophys. Res., 103, 32,119-32,128, 1998.*

*Andreae, M. O., and Merlet, P., Emission of trace gases and aerosols from biomass burning: Global Biogeochemical Cycles, 15, 955-966, 2001.*

*Artaxo P., Rizzo, L. V., Brito, J. F., Barbosa, H. M. J., Arana A., Sena E. T., Cirino G. G., Bastos W., Martins S. T., and Andreae M. O.: Atmospheric aerosol in Amazonia and land use change: from natural biogenic to biomass burning conditions. Faraday Discuss., 165, 203-235, doi:10.1039/C3FD00052D, 2013.*

*Ben-Ami, Y., Koren, I., Rudich, Y., Artaxo, P., Martin, S. T., and Andreae, M. O.: Transport of North African dust from the Bodélé depression to the Amazon Basin: a case study, Atmos. Chem. Phys., 10, 7533-7544, doi:10.5194/acp-10-7533-2010, 2010.*

*Bergstrom, R. W., Pilewskie, P., Russell, P. B., Redemann, J., Bond, T. C., Quinn, P. K., & Sierau, B. (2007). Spectral absorption properties of atmospheric aerosols. Atmospheric Chemistry and Physics Discussions, 7(4), 10669–10686. http://doi.org/10.5194/acpd-7-10669-2007*

*Cwiertny, D. M., Baltrusaitis, J., Hunter, G. J., Laskin, A., Scherer, M. M., Grassian, V. H.: Characterization and acid-mobilization study of iron-containing mineral dust source materials. J. Geophys. Res., 113, D05202, doi:10.1029/2007JD009332, 2008.*

*Eck, T. F., Holben, B. N., Reid, J. S., Dubovik, O., Smirnov, A., O'Neill, N. T., Slutsker, I., and Kinne, S.: Wavelength dependence of the optical depth of biomass burning, urban, and desert dust aerosols, J. Geophys. Res., 104, 31333–31349, doi:10.1029/1999jd900923, 1999.*

*Gaudichet, A., Echalar, F., Chatenet, B., Quisefit, J. P., Malingre, G., Cachier, H., Buat-Ménard, P., Artaxo, P., and Maenhaut, W., Trace elements in tropical African savanna biomass burning aerosols: J. Atmos. Chem., 22, 19-39, 1995.*

*Lack, D. A., & Langridge, J. M. (2013). On the attribution of black and brown carbon light absorption using the Angstrom exponent. Atmospheric Chemistry and Physics, 13(20), 10535–10543. http://doi.org/10.5194/acp-13-10535-2013.*

*Longo, A. F., Feng, Y., Lai, B., Landing, W. M., Shelley, R. U., Nenes, A., Mihalopou-los, N., Violaki, K., Ingall, E. D.: Influence of Atmospheric Processes on the Solubil-ity and Composition of Iron in Saharan Dust. Environ. Sci. Technol., 50 (13), 6912-6920, doi: 10.1021/acs.est.6b02605, 2016.*

*Maenhaut, W., Salma, I., Cafmeyer, J., Annegarn, H. J., and Andreae, M. O., Regional atmospheric aerosol composition and sources in the Eastern Transvaal, South Africa, and impact of biomass burning: J. Geophys. Res., 101, 23,631-23,650, 1996.*

*Ogunjobi, K.O., He, Z., & Simmer, C.: Spectral aerosol optical properties from AERONET Sunphotometric measurements over West Africa, Atmos. Res., 88, 89-107, 2008.*

*Pauliquevis, T., Lara, L. L., Antunes, M. L., and Artaxo, P.: Aerosol and precipitation chemistry measurements in a remote site in Central Amazonia: the role of biogenic contribution. Atmos.*

*Chem. Phys., 12, 4987-5015, doi:10.5194/acp-12-4987, 2012.*

*Swap, R., Garstang, M., Greco, S. Talbot, R., and Kållberg, P.: Saharan dust in the Am-azon Basin. Tellus, 44, 133-149, 1992.*

*Talbot, R. W., Andreae, M. O., Berresheim, H., Artaxo, P., Garstang, M., Harriss, R. C., Beecher, K. M., and Li, S. M.: Aerosol chemistry during the wet season in Central Amazonia: The influence of long-range transport: J. Geophys. Res., 95, 16,955-16,969, 1990.*

*Yu, H., Chin, M., Yuan, T., Bian, H., Remer, L. A., Prospero, J. M., Omar, A., Winker, D., Yang, Y., Zhang, Y., Zhang, Z., and Zhao, C.: The fertilizing role of African dust in the Amazon rainforest: A first multiyear assessment based on data from Cloud-Aerosol Lidar and Infrared Pathfinder Satellite Observations. Geophys. Res. Lett., 42, 1984-1991, doi:10.1002/2015GL063040, 2015.*

*Zhu, X. R., Prospero, J. M., and Millero, F. J.: Diel variability of soluble Fe(II) and sol-uble total Fe in North African dust in the trade winds at Barbados. J. Geophys. Res., 102, 21297-21305, doi:10.1029/97JD01313 , 1997.*

---

## Author Comment (AC2) · 14 Jan 2017

1. The problem of this manuscript that begins by the Title and the Abstract is the generalization of ideas. First I suggest that the authors be more specific in the Title, it should focus on the soluble fraction Fe(II)/Fe(III) issue, which is something important but not enough to account a full story about the Amazon rainforest ecology. R: Yes, we agree with the reviewer, and the title was changed to: "Soluble iron nutrients in Saharan dust over the central Amazon rainforest" to match better with the main goal. The text was also restructured to emphasize the iron soluble fraction and the frequent long-range transport of African aerosols.

2. In the introduction, clarify at Line 124 (pag 6) when the authors say ": Considering

that iron is absorbed by plants only as soluble Fe(II)/Fe(III)", previously the authors have stated that Fe(III) was also recovered by the action of rhizosphere. R: The sentence was confusing as indicated by the reviewer. Plants require the micronutrient iron in small amounts and the absorption can vary according to the species. Iron uptake can be as Fe (II) or (III) and the absorption fraction depends on the ability of the plant to reduce it to Fe (II). In this role, the pH is important for solubility and therefore the iron availability. The text has been changed accordingly.

3. In methods, Line 142, pag 6, provide filter porosity. R: The information was added to the text as follows: (L. 111-112)"Atmospheric particles were collected on Nuclepore® polycarbonate filters (47 mm diameter, 0.8 $\mu$m pore size, Whatman® Nuclepore)"

4. In Item 2.5, specify how the samples were storage and if the observations were conducted in the filed or in a particular laboratory condition? R: After sampling the filters were stored at 4°C in the field and then carried to a laboratory to perform the ion chromatography analysis. The information was added to the text: (L. 117-118) "After sampling, the filters were immediately stored in sterile flasks under refrigeration until laboratory analysis."

5. The sentence in Line 229-231 "The mass concentration of particles over the Amazon Basin in the wet season is typically around 10 ug m-3 in locations that are influenced by biomass burning emissions", : : : here there is confusion on the wet season and biomass burning season. Which reference is attributed to the mass concentration mentioned? R: The reviewer is right, the information was confusing. The text was rewritten as follows: (L. 297-303)" The mass concentration of PM10 particles in Amazonia is close to background in most areas throughout the basin during the wet season. Central Amazonia is characterized by a weak influence of anthropogenic emissions and aerosol mass concentrations are low during the wet season - typically 7 $\mu$g m Ìǔ 3; even the most impacted areas do not exceed 10 $\mu$g m Ìǔ 3 due to intensive rain and the corresponding inhibition of biomass burning (Artaxo et al., 2002; Artaxo et al. 2013;

Martin et al., 2010). Increased mass concentrations may occur due to African dust events that reach the Amazon forest in this season (Talbot et al., 1990; Martin et al., 2010)."

6. Part of the text in Lines 254 to 268 (pags 11-12) could be placed at Material and Methods. If possible the authors could place BCe time series superimposed to mass concentrations at Figure 1. This could give some idea on the contribution of BC to the bulk atmospheric concentrations, or if they are lagged in time. R: Following the referee's suggestion, we moved part of the text from the Results to the Material and Methods section. Part of section 2.4 was reformulated as follows: (L. 160-168) "Equivalent black carbon concentrations (BCe) were obtained by a Multi Angle Absorption Photometer (MAAP, Model 5012, Thermo Electron Group, USA; $\lambda$ = 670 nm), based on light absorption measurements at 637 nm. An absorption cross section value of 6.6 m2 g was used for the conversion of measured absorption coefficients into BCe concentrations (Petzold et al., 2005). Soot, mineral dust, and biogenic particles are light absorbers (Moosmüller et al., 2009; 2011; Guyon et al., 2004; Andreae and Gelencsér 2006) and may contribute to the observed BCe signal. The relative contributions of particle sources to BCe can be investigated by considering the absorption spectral variability, by means of the so called Absorption Ångström Exponent (AAE)." The information was also added in section 3.2: (L. 316-317) "The concentrations of black carbon equivalent (BCe) measured online during this intensive campaign represented on average 1.5% of PM10 mass concentrations, ranging from 0 to 0.3 $\mu$g m Ĭ $^3$." (L. 321-323) "Figure 2 and Table 1 show that BCe concentrations significantly increased regionally during 1-8 April, coinciding with the increase in PM10 and particle soluble fraction concentrations"

7. In Table 2, how the elemental analysis was conducted for Cu, Zn, Na, Ca, K and Mg? and about the NH4 ? R: The experimental details were included in the Methods section as follows:(L. 140-146) "For the cation analysis, ultrapure water and methanesulfonic acid (MSA) was used as the eluent at a 20 mM constant concentration, with automatic

suppression (CSRS suppressor - 2 mm), and with a 0.33 mL min Ìű 1 system flow through an IonPac CG-12 guard column (2 x 50 mm) and CS-12 (2 x 250 mm) capillary column. This resulted in a 14 min running time for each injection. For soluble Na, NH4, K, Mg and Ca the detection limits (USEPA, 1997) were 2.0, 1.3, 0.9, 0.7, and 1.8 $\mu$g L Ìű 1, respectively, and the expanded uncertainties at the 95% level of confidence (BIPM, 2008) were of 9, 7, 21, 11, and 23 %, respectively."

8. In the title of Table 2, it is not "aerosol characterization" it is aerosol composition; it does not correspond to "during the Saharan dust event ", it is before, along and after the event. R: The reviewer is correct. Table 2 was removed from the text, and essential information was added to Figure 2 and in the text.

9. In Lines 279-282 the authors say that K, Zn and Cu are of biogenic sources, probably mostly emitted during biomass burning. If the detected pulse of dust in this work is coincident with an African biomass burning event as pointed by the authors, what is the level of certainty to say that their main source is the mineral fraction? R: Biomass burning in Africa could have contributed some Fe, but unfortunately, little is known about Fe emissions from savanna fires, and the available data span a wide range. From the work of Gaudichet et al. (1995), one can derive an Fe content of 0.016% in savanna smoke TPM, which with a peak biomass smoke concentration of 4 $\mu$g m-3 would only give 0.6 ng Fe m 3. Using the BC/Fe ratio of ca. 40 from Maenhaut et al. (1996) and the peak BCe concentration of 0.3 $\mu$g m-3, we can estimate ca. 8 ng Fe m-3. Finally, using the Fe emission factor of 0.026 g/kg for African savanna fires from Andreae et al. (1998) and the BC emission factor of 0.6 g/kg from Andreae and Merlet (2001 and updates), we can estimate a peak pyrogenic Fe contribution of 13 ng m-3.This compares to 64 ng m-3 of soluble iron at the same time, and given that only a small fraction of the Fe in biomass smoke is likely to be soluble, it is clear that the dominant fraction of soluble Fe comes from the African mineral dust. Discussion on this issue has been added in Section 3.1.

10. In Line 322, the comparison of the present work with Andreae et al. (2015): does

both work have same methods and associated errors? Results of Andreae et al. (2015) correspond to what period of the year. Specify please. R: The results of Andreae et al. (2015) correspond to the period from 7 March to 21 April 2012, and the chromatography analyses have the same method and associated errors. The information was added to the text as follows: (L. 373-375)" The soluble Fe(III) concentrations were significantly higher than those reported by Andreae et al. (2015) from earlier measurements at the same site, which had also been made during the wet season and using the same quantification method."

11. Text in lines 333-338 is unnecessary. R: We agree with the reviewer, the text was removed.

12. Dates in Figure 2 is unreadable. R: We agree with the reviewer, the Figure 2 was replaced by another with readable information.

13. Figure 3 should be completely edited. It is not possible to use the Hysplit output directly. For Figure 3, use ensembles, not a single trajectory; a family of trajectories gives a better idea of all geographical contributions. R: We agree with the reviewer and the figure was edited as requested, showing the backward trajectories to illustrate the intercontinental transport.

15. Lines 369-374; Figure 1 shows before, along and after the "dust storm", I suggest that the authors run the Hysplit model in these 3 circumstances and then make their conclusions. R: We agree with the reviewer. Figure 1 was replaced and comments changed with new conclusions added according to the suggestion.

16. In Line 387 provide complete localization of the three AERONET sites: Dakar and Ilorin in Africa, and Embrapa/ Manaus in Amazon. R: The geographical coordinates of these AERONET sites have been included for the AERONET sites (L. 228): Dakar (14° 23' 38"N; 16° 57' 32"W) and Ilorin (08° 19' 12"N; 04° 20' 24"E) in Africa, and Embrapa/Manaus (02° 53' 12"S; 59° 58' 12"W) in Amazonia (L. 804).

17. In Figure 5, AOD do not distinguish dust from biomass burning products. From the location of higher AODs in the diagrams it seems that your source could have some contribution from biomass burning than mineral dust. Also the results presented in the Hysplit are not totally in accordance to wind flows at the charts at Figure 5. Maybe the source is a net combination of both; I strongly suggest that the authors add a map with fire spots for the period of sampling, so to make better differentiate. R: Yes, there is clearly a contribution from biomass burning. To clarify the possible contribution of smoke, we added in Figure 6 fire spots observed during the sampling period in both continents, South America and Africa. Over South America, major fire spots areas (Brazilian cerrado ecosystem and the north portion of the continent) are not upwind of the ATTO site, which reduces the site exposure to smoke plumes from these principal regional spots. In Africa, the main fire spots areas are downwind of the Sahara desert, along the west coast of Africa, therefore on the way of the dust flux toward the Tropical Atlantic and South America, which could promote transport of a mixture of smoke and dust. The referee is right, the AOD does not distinguish dust from biomass burning. Thus, observing exclusively the AOD map it is hard to say which is one dominant, dust or smoke. However, from the analysis of the Angstrom Exponent (AE) against AOD measured using data from AERONET sites located in the Sub-Saharan areas with high AOD (Ilorin, Dakar and Cape Verde) it is possible to assess the dominant aerosol type across west Africa. The AE is close to zero when aerosol plumes are dominated by large particles (e.g., sea salt, soil dust, biogenic) and higher than 1.0 when fine particles (e.g., from biomass burning and fossil fuel combustion) are dominant (Eck et al., 1999). It is well established that an increase in AOD associated with a decrease in AE in the sub-Saharan region is associated with the presence of dust plumes, and the opposite, increase in AE associated with an AOD increase is related to biomass burning plumes (Ogunjobi et al., 2008, Eck et al.1999). Although a contribution from biomass burning smoke is very likely in these areas, the plots of AE against AOD for Dakar, Cape Verde, and particularly Ilorin, during the four periods analyzed in Figure 6 shows that dust plumes clearly dominated during the higher AOD scenarios. The plot

for the Ilorin case was included as an example in the manuscript to corroborate that the plume that left Africa towards the Tropical Atlantic and South America was dominated by dust aerosols. The same analysis performed for the AERONET station located in central Amazonia (northwest of Manaus) also suggested that regional AOD increases during the sampling period were dominantly connected with decreases in AE, and thus increased coarse mode particles. This is consistent with Castro Videla et al. (2013), who showed that peaks on AOD in Central Amazonia during the wet season had a significant contribution from coarse mode particles. As discussed in the response to Comment 2 of Reviewer 1, a TPM contribution of about 20% from biomass burning can be estimated.

18. In Figure 6, what is MC? Please, correct the legend of time. R: MC stands for Mass Concentration. The legend was corrected as indicated.

19. The discussion on fungi is very poor. There is none description of the species nor anything on their biogeography. The lesson of this result is the fact that a more detail aerobiological research should be conducted to be published. R: The fungi identification underscores long distance transport, but doesn't allude to a specific site. We included our observations of coarse particles during the dust event to see if there were likely to be readily identifiable inputs from the canopy that might add to the iron analysis. Bioaerosol identification would also help confirm if any coarse particles that were mixed with the dust were of other than local origin. The spores identified in the samples do not add soluble iron to the analyzed extracts.

20. In Line 463-465, the authors say "Smoke plumes are known to entrain fungi over long distances (Mims and Mims, 2004). Dust from Lake Chad is rich in bacteria and fungi." Here becomes explicitly that the authors are not able to stablish a source of the particulate matter entering Amazon in the considered event: Saharan mineral dust or sub-Saharan biomass burning? R: The particles found have influence from the plumes originated from the African continent as confirmed by trajectories. The long distance transport is evidence from our findings but we cannot be more precise about the source

of the fungi without further analysis.

21. The Amazon itself is a fantastic source of bacteria and fungi, and only an endemic specie of Africa, detected in Amazon, at high level (ex. The top of the ATTO) could make a clear distinction. R: The reviewer is correct. We cannot fully compare the bioaerosol results because previous studies cultured air samples of viable spores only, and analysed with high throughput sequencing. Only a few types of fungi were detected at the species level.

22. In item 3.5 the authors says that "a small amount of atmospheric iron could affect the microbiota in the canopy, rather than have a significant effect on soil and root uptake for plants." This is an speculation and from this work it is not possible to conclude anything. R: Yes, we agree with the reviewer. The sentence was removed and the section restructured to emphasize our finding.

23. In my opinion, most of item 3.5 is Introduction to the study since most of the text is compilation from the literature associated to this work. R: Yes, we agree with the reviewer. Some parts of section 3.5 were placed in the Introduction and most of the section was rewritten (L. 439-458).

24. The conclusion unrealistic, should be reduced to the basic findings. R: The conclusion was rewritten to focus on our findings (L. 460-467).

Please also note the supplement to this comment:
http://www.atmos-chem-phys-discuss.net/acp-2016-557/acp-2016-557-AC2-supplement.pdf

**Supplement:**

*Response to Referee #2*

**Abstract**
**1. The problem of this manuscript that begins by the Title and the Abstract is the generalization of ideas. First I suggest that the authors be more specific in the Title, it should focus on the soluble fraction Fe(II)/Fe(III) issue, which is something important but not enough to account a full story about the Amazon rainforest ecology.**
*R: Yes, we agree with the reviewer, and the title was changed to: "Soluble iron nutrients in Saharan dust over the central Amazon rainforest" to match better with the main goal. The text was also restructured to emphasize the iron soluble fraction and the frequent long-range transport of African aerosols.*

**2. In the introduction, clarify at Line 124 (pag 6) when the authors say ": Considering that iron is absorbed by plants only as soluble Fe(II)/Fe(III)", previously the authors have stated that Fe(III) was also recovered by the action of rhizosphere.**
*R: The sentence was confusing as indicated by the reviewer. Plants require the micronutrient iron in small amounts and the absorption can vary according to the species. Iron uptake can be as Fe (II) or (III) and the absorption fraction depends on the ability of the plant to reduce it to Fe (II). In this role, the pH is important for solubility and therefore the iron availability. The text has been changed accordingly.*

**3. In methods, Line 142, pag 6, provide filter porosity.**
*R: The information was added to the text as follows: (L. 111-112)"Atmospheric particles were collected on Nuclepore® polycarbonate filters (47 mm diameter, 0.8 μm pore size, Whatman® Nuclepore)"*

**4. In Item 2.5, specify how the samples were storage and if the observations were conducted in the filed or in a particular laboratory condition?**
*R: After sampling the filters were stored at 4°C in the field and then carried to a laboratory to perform the ion chromatography analysis. The information was added to the text: (L. 117-118) "After sampling, the filters were immediately stored in sterile flasks under refrigeration until laboratory analysis."*

**5. The sentence in Line 229-231 "The mass concentration of particles over the Amazon Basin in the wet season is typically around 10 ug m-3 in locations that are influenced by biomass burning emissions", : : : here there is confusion on the wet season and biomass burning season. Which reference is attributed to the mass concentration mentioned?**
*R: The reviewer is right, the information was confusing. The text was rewritten as follows: (L. 297-303)" The mass concentration of PM10 particles in Amazonia is close to background in most areas throughout the basin during the wet season. Central Amazonia is characterized by a weak influence of anthropogenic emissions and aerosol mass concentrations are low during the wet season - typically 7 μg m−3; even the most impacted areas do not exceed 10 μg m−3 due to intensive rain and the corresponding inhibition of biomass burning (Artaxo et al., 2002; Artaxo et al. 2013; Martin et al., 2010). Increased mass concentrations may occur due to African dust events that reach the Amazon forest in this season (Talbot et al., 1990; Martin et al., 2010)."*

**6. Part of the text in Lines 254 to 268 (pags 11-12) could be placed at Material and Methods. If possible the authors could place BCe time series superimposed to mass concentrations at Figure 1. This could give some idea on the contribution of BC to the**

**bulk atmospheric concentrations, or if they are lagged in time.**

*R: Following the referee's suggestion, we moved part of the text from the Results to the Material and Methods section. Part of section 2.4 was reformulated as follows:*

*(L. 160-168) "Equivalent black carbon concentrations (BCe) were obtained by a Multi Angle Absorption Photometer (MAAP, Model 5012, Thermo Electron Group, USA; $\lambda$ = 670 nm), based on light absorption measurements at 637 nm. An absorption cross section value of 6.6 m2 g was used for the conversion of measured absorption coefficients into BCe concentrations (Petzold et al., 2005). Soot, mineral dust, and biogenic particles are light absorbers (Moosmüller et al., 2009; 2011; Guyon et al., 2004; Andreae and Gelencsér 2006) and may contribute to the observed BCe signal. The relative contributions of particle sources to BCe can be investigated by considering the absorption spectral variability, by means of the so called Absorption Ångström Exponent (AAE)."*

*The information was also added in section 3.2:*

*(L. 316-317) "The concentrations of black carbon equivalent (BCe) measured online during this intensive campaign represented on average 1.5% of PM10 mass concentrations, ranging from 0 to 0.3 µg m$^{-3}$." (L. 321-323) "Figure 2 and Table 1 show that BCe concentrations significantly increased regionally during 1-8 April, coinciding with the increase in PM10 and particle soluble fraction concentrations"*

**7. In Table 2, how the elemental analysis was conducted for Cu, Zn, Na, Ca, K and Mg? and about the NH4 ?**

*R: The experimental details were included in the Methods section as follows:(L. 140-146) "For the cation analysis, ultrapure water and methanesulfonic acid (MSA) was used as the eluent at a 20 mM constant concentration, with automatic suppression (CSRS suppressor - 2 mm), and with a 0.33 mL min$^{-1}$ system flow through an IonPac CG-12 guard column (2 x 50 mm) and CS-12 (2 x 250 mm) capillary column. This resulted in a 14 min running time for each injection. For soluble Na, NH4, K, Mg and Ca the detection limits (USEPA, 1997) were 2.0, 1.3, 0.9, 0.7, and 1.8 µg L$^{-1}$, respectively, and the expanded uncertainties at the 95% level of confidence (BIPM, 2008) were of 9, 7, 21, 11, and 23 %, respectively."*

**8. In the title of Table 2, it is not "aerosol characterization" it is aerosol composition; it does not correspond to "during the Saharan dust event ", it is before, along and after the event.**

*R: The reviewer is correct. Table 2 was removed from the text, and essential information was added to Figure 2 and in the text.*

**9. In Lines 279-282 the authors say that K, Zn and Cu are of biogenic sources, probably mostly emitted during biomass burning. If the detected pulse of dust in this work is coincident with an African biomass burning event as pointed by the authors, what is the level of certainty to say that their main source is the mineral fraction?**

*R: Biomass burning in Africa could have contributed some Fe, but unfortunately, little is known about Fe emissions from savanna fires, and the available data span a wide range. From the work of Gaudichet et al. (1995), one can derive an Fe content of 0.016% in savanna smoke TPM, which with a peak biomass smoke concentration of 4 µg m$^{-3}$ would only give 0.6 ng Fe m$^{-3}$. Using the BC/Fe ratio of ca. 40 from Maenhaut et al. (1996) and the peak BCe concentration of 0.3 µg m$^{-3}$, we can estimate ca. 8 ng Fe m$^{-3}$. Finally, using the Fe emission factor of 0.026 g/kg for African savanna fires from Andreae et al. (1998) and the BC emission factor of 0.6 g/kg from Andreae and Merlet (2001 and updates), we can estimate a peak pyrogenic Fe contribution of 13 ng m$^{-3}$.This compares to 64 ng m$^{-3}$ of soluble iron at the same time, and given that only a small fraction of the Fe in biomass smoke is likely to be soluble, it is clear that the dominant fraction of soluble Fe comes from the African mineral dust.*

*Discussion on this issue has been added in Section 3.1.*

**10. In Line 322, the comparison of the present work with Andreae et al. (2015): does both work have same methods and associated errors? Results of Andreae et al. (2015) correspond to what period of the year. Specify please.**
*R: The results of Andreae et al. (2015) correspond to the period from 7 March to 21 April 2012, and the chromatography analyses have the same method and associated errors. The information was added to the text as follows: (L. 373-375)" The soluble Fe(III) concentrations were significantly higher than those reported by Andreae et al. (2015) from earlier measurements at the same site, which had also been made during the wet season and using the same quantification method."*

**11. Text in lines 333-338 is unnecessary.**
*R: We agree with the reviewer, the text was removed.*

**12. Dates in Figure 2 is unreadable.**
*R: We agree with the reviewer, the Figure 2 was replaced by another with readable information.*

**13. Figure 3 should be completely edited. It is not possible to use the Hysplit output directly. For Figure 3, use ensembles, not a single trajectory; a family of trajectories gives a better idea of all geographical contributions.**
*R: We agree with the reviewer and the figure was edited as requested, showing the backward trajectories to illustrate the intercontinental transport.*

**15. Lines 369-374; Figure 1 shows before, along and after the "dust storm", I suggest that the authors run the Hysplit model in these 3 circumstances and then make their conclusions.**
*R: We agree with the reviewer. Figure 1 was replaced and comments changed with new conclusions added according to the suggestion.*

**16. In Line 387 provide complete localization of the three AERONET sites: Dakar and Ilorin in Africa, and Embrapa/ Manaus in Amazon.**
*R: The geographical coordinates of these AERONET sites have been included for the AERONET sites (L. 228): Dakar (14° 23' 38''N; 16° 57' 32''W) and Ilorin (08° 19' 12''N; 04° 20' 24''E) in Africa, and Embrapa/Manaus (02° 53' 12''S; 59° 58' 12''W) in Amazonia (L. 804).*

**17. In Figure 5, AOD do not distinguish dust from biomass burning products. From the location of higher AODs in the diagrams it seems that your source could have some contribution from biomass burning than mineral dust. Also the results presented in the Hysplit are not totally in accordance to wind flows at the charts at Figure 5. Maybe the source is a net combination of both; I strongly suggest that the authors add a map with fire spots for the period of sampling, so to make better differentiate.**
*R: Yes, there is clearly a contribution from biomass burning. To clarify the possible contribution of smoke, we added in Figure 6 fire spots observed during the sampling period in both continents, South America and Africa. Over South America, major fire spots areas (Brazilian cerrado ecosystem and the north portion of the continent) are not upwind of the ATTO site, which reduces the site exposure to smoke plumes from these principal regional*

*spots. In Africa, the main fire spots areas are downwind of the Sahara desert, along the west coast of Africa, therefore on the way of the dust flux toward the Tropical Atlantic and South America, which could promote transport of a mixture of smoke and dust. The referee is right, the AOD does not distinguish dust from biomass burning. Thus, observing exclusively the AOD map it is hard to say which is one dominant, dust or smoke. However, from the analysis of the Angstrom Exponent (AE) against AOD measured using data from AERONET sites located in the Sub-Saharan areas with high AOD (Ilorin, Dakar and Cape Verde) it is possible to assess the dominant aerosol type across west Africa.  The AE is close to zero when aerosol plumes are dominated by large particles (e.g., sea salt, soil dust, biogenic) and higher than 1.0 when fine particles (e.g., from biomass burning and fossil fuel combustion) are dominant (Eck et al., 1999). It is well established that an increase in AOD associated with a decrease in AE in the sub-Saharan region is associated with the presence of dust plumes, and the opposite, increase in AE associated with an AOD increase is related to biomass burning plumes (Ogunjobi et al., 2008, Eck et al.1999).  Although a contribution from biomass burning smoke is very likely in these areas, the plots of AE against AOD for Dakar, Cape Verde, and particularly Ilorin, during the four periods analyzed in Figure 6 shows that dust plumes clearly dominated during the higher AOD scenarios. The plot for the Ilorin case was included as an example in the manuscript to corroborate that the plume that left Africa towards the Tropical Atlantic and South America was dominated by dust aerosols. The same analysis performed for the AERONET station located in central Amazonia (northwest of Manaus) also suggested that regional AOD increases during the sampling period were dominantly connected with decreases in AE, and thus increased coarse mode particles. This is consistent with Castro Videla et al. (2013), who showed that peaks on AOD in Central Amazonia during the wet season had a significant contribution from coarse mode particles. As discussed in the response to Comment 2 of Reviewer 1, a TPM contribution of about 20% from biomass burning can be estimated.*

**18. In Figure 6, what is MC? Please, correct the legend of time**.
*R: MC stands for Mass Concentration. The legend was corrected as indicated.*

**19. The discussion on fungi is very poor. There is none description of the species nor anything on their biogeography. The lesson of this result is the fact that a more detail aerobiological research should be conducted to be published.**
*R: The fungi identification underscores long distance transport, but doesn't allude to a specific site. We included our observations of coarse particles during the dust event to see if there were likely to be readily identifiable inputs from the canopy that might add to the iron analysis. Bioaerosol identification would also help confirm if any coarse particles that were mixed with the dust were of other than local origin. The spores identified in the samples do not add soluble iron to the analyzed extracts.*

**20. In Line 463-465, the authors say "Smoke plumes are known to entrain fungi over long distances (Mims and Mims, 2004). Dust from Lake Chad is rich in bacteria and fungi." Here becomes explicitly that the authors are not able to stablish a source of the particulate matter entering Amazon in the considered event: Saharan mineral dust or sub-Saharan biomass burning?**
*R: The particles found have influence from the plumes originated from the African continent as confirmed by trajectories. The long distance transport is evidence from our findings but we cannot be more precise about the source of the fungi without further analysis.*

**21. The Amazon itself is a fantastic source of bacteria and fungi, and only an endemic specie of Africa, detected in Amazon, at high level (ex. The top of the ATTO) could make a clear distinction.**

*R: The reviewer is correct. We cannot fully compare the bioaerosol results because previous studies cultured air samples of viable spores only, and analysed with high throughput sequencing. Only a few types of fungi were detected at the species level.*

**22. In item 3.5 the authors says that "a small amount of atmospheric iron could affect the microbiota in the canopy, rather than have a significant effect on soil and root uptake for plants." This is an speculation and from this work it is not possible to conclude anything.**
*R: Yes, we agree with the reviewer. The sentence was removed and the section restructured to emphasize our finding.*

**23. In my opinion, most of item 3.5 is Introduction to the study since most of the text is compilation from the literature associated to this work.**
*R: Yes, we agree with the reviewer. Some parts of section 3.5 were placed in the Introduction and most of the section was rewritten (L. 439-458).*

**24. The conclusion unrealistic, should be reduced to the basic findings.**
*R: The conclusion was rewritten to focus on our findings (L. 460-467).*

*References:*

*Andreae, M. O., Andreae, T. W., Annegarn, H., Beer, F., Cachier, H., Elbert, W., Harris, G. W., Maenhaut, W., Salma, I., Swap, R., Wienhold, F. G., and Zenker, T., Airborne studies of aerosol emissions from savanna fires in southern Africa: 2. Aerosol chemical composition: J. Geophys. Res., 103, 32,119-32,128, 1998.*

*Andreae, M. O., and Merlet, P., Emission of trace gases and aerosols from biomass burning: Global Biogeochemical Cycles, 15, 955-966, 2001.*

*Artaxo P., Rizzo, L. V., Brito, J. F., Barbosa, H. M. J., Arana A., Sena E. T., Cirino G. G., Bastos W., Martins S. T., and Andreae M. O.: Atmospheric aerosol in Amazonia and land use change: from natural biogenic to biomass burning conditions. Faraday Discuss., 165, 203-235, doi:10.1039/C3FD00052D, 2013.*

*Ben-Ami, Y., Koren, I., Rudich, Y., Artaxo, P., Martin, S. T., and Andreae, M. O.: Transport of North African dust from the Bodélé depression to the Amazon Basin: a case study, Atmos. Chem. Phys., 10, 7533-7544, doi:10.5194/acp-10-7533-2010, 2010.*

*Bergstrom, R. W., Pilewskie, P., Russell, P. B., Redemann, J., Bond, T. C., Quinn, P. K., & Sierau, B. (2007). Spectral absorption properties of atmospheric aerosols. Atmospheric Chemistry and Physics Discussions, 7(4), 10669–10686. http://doi.org/10.5194/acpd-7-10669-2007*

*Cwiertny, D. M., Baltrusaitis, J., Hunter, G. J., Laskin, A., Scherer, M. M., Grassian, V. H.: Characterization and acid-mobilization study of iron-containing mineral dust source materials. J. Geophys. Res., 113, D05202, doi:10.1029/2007JD009332, 2008.*

*Eck, T. F., Holben, B. N., Reid, J. S., Dubovik, O., Smirnov, A., O'Neill, N. T., Slutsker, I., and Kinne, S.: Wavelength dependence of the optical depth of biomass burning, urban, and desert dust aerosols, J. Geophys. Res., 104, 31333–31349, doi:10.1029/1999jd900923, 1999.*

*Gaudichet, A., Echalar, F., Chatenet, B., Quisefit, J. P., Malingre, G., Cachier, H., Buat-Ménard, P., Artaxo, P., and Maenhaut, W., Trace elements in tropical African savanna biomass burning aerosols: J. Atmos. Chem., 22, 19-39, 1995.*

*Lack, D. A., & Langridge, J. M. (2013). On the attribution of black and brown carbon light absorption using the Angstrom exponent. Atmospheric Chemistry and Physics, 13(20), 10535–10543. http://doi.org/10.5194/acp-13-10535-2013.*

*Longo, A. F., Feng, Y., Lai, B., Landing, W. M., Shelley, R. U., Nenes, A., Mihalopou-los, N., Violaki, K., Ingall, E. D.: Influence of Atmospheric Processes on the Solubil-ity and Composition of Iron in Saharan Dust. Environ. Sci. Technol., 50 (13), 6912-6920, doi: 10.1021/acs.est.6b02605, 2016.*

*Maenhaut, W., Salma, I., Cafmeyer, J., Annegarn, H. J., and Andreae, M. O., Regional atmospheric aerosol composition and sources in the Eastern Transvaal, South Africa, and impact of biomass burning: J. Geophys. Res., 101, 23,631-23,650, 1996.*

*Ogunjobi, K.O., He, Z., & Simmer, C.: Spectral aerosol optical properties from AERONET Sunphotometric measurements over West Africa, Atmos. Res., 88, 89-107, 2008.*

*Pauliquevis, T., Lara, L. L., Antunes, M. L., and Artaxo, P.: Aerosol and precipitation chemistry measurements in a remote site in Central Amazonia: the role of biogenic contribution. Atmos.*

*Chem. Phys., 12, 4987-5015, doi:10.5194/acp-12-4987, 2012.*

*Swap, R., Garstang, M., Greco, S. Talbot, R., and Kållberg, P.: Saharan dust in the Am-azon Basin. Tellus, 44, 133-149, 1992.*

*Talbot, R. W., Andreae, M. O., Berresheim, H., Artaxo, P., Garstang, M., Harriss, R. C., Beecher, K. M., and Li, S. M.: Aerosol chemistry during the wet season in Central Amazonia: The influence of long-range transport: J. Geophys. Res., 95, 16,955-16,969, 1990.*

*Yu, H., Chin, M., Yuan, T., Bian, H., Remer, L. A., Prospero, J. M., Omar, A., Winker, D., Yang, Y., Zhang, Y., Zhang, Z., and Zhao, C.: The fertilizing role of African dust in the Amazon rainforest: A first multiyear assessment based on data from Cloud-Aerosol Lidar and Infrared Pathfinder Satellite Observations. Geophys. Res. Lett., 42, 1984-1991, doi:10.1002/2015GL063040, 2015.*

*Zhu, X. R., Prospero, J. M., and Millero, F. J.: Diel variability of soluble Fe(II) and sol-uble total Fe in North African dust in the trade winds at Barbados. J. Geophys. Res., 102, 21297-21305, doi:10.1029/97JD01313 , 1997.*